# Biophysical models reveal the relative importance of transporter proteins and impermeant anions in chloride homeostasis

Kira M Düsterwald[1,2,3], Christopher B Currin[1,2,3†], Richard J Burman[1,2,3†], Colin J Akerman[4], Alan R Kay[5], Joseph V Raimondo[1,2,3*]

[1]Division of Cell Biology, Department of Human Biology, Faculty of Health Sciences, University of Cape Town, Cape Town, South Africa; [2]Neuroscience Institute, University of Cape Town, Cape Town, South Africa; [3]Institute of Infectious Disease and Molecular Medicine, University of Cape Town, Cape Town, South Africa; [4]Department of Pharmacology, University of Oxford, Oxford, United Kingdom; [5]Department of Biology, University of Iowa, Iowa City Iowa, United States

**\*For correspondence:**
joseph.raimondo@uct.ac.za

[†]These authors contributed equally to this work

**Competing interests:** The authors declare that no competing interests exist.

**Abstract** Fast synaptic inhibition in the nervous system depends on the transmembrane flux of $Cl^-$ ions based on the neuronal $Cl^-$ driving force. Established theories regarding the determinants of $Cl^-$ driving force have recently been questioned. Here, we present biophysical models of $Cl^-$ homeostasis using the pump-leak model. Using numerical and novel analytic solutions, we demonstrate that the $Na^+/K^+$-ATPase, ion conductances, impermeant anions, electrodiffusion, water fluxes and cation-chloride cotransporters (CCCs) play roles in setting the $Cl^-$ driving force. Our models, together with experimental validation, show that while impermeant anions can contribute to setting $[Cl^-]_i$ in neurons, they have a negligible effect on the driving force for $Cl^-$ locally and cell-wide. In contrast, we demonstrate that CCCs are well-suited for modulating $Cl^-$ driving force and hence inhibitory signaling in neurons. Our findings reconcile recent experimental findings and provide a framework for understanding the interplay of different chloride regulatory processes in neurons.
DOI: https://doi.org/10.7554/eLife.39575.001

## Introduction

Fast synaptic inhibition in the nervous system is mediated by type A γ-aminobutyric acid receptors ($GABA_A$Rs) and glycine receptors (GlyRs), which are primarily permeable to chloride ($Cl^-$) (*Farrant and Kaila, 2007*). Together with the neuronal membrane potential, the transmembrane gradient for $Cl^-$ sets the driving force for $Cl^-$ flux across these receptors, controlling the properties of inhibitory signaling. Modification of neuronal intracellular $Cl^-$ concentration ($[Cl^-]_i$) has been shown to play a causative role in multiple neurological diseases including epilepsy, chronic pain, schizophrenia and autism (*Rivera et al., 2004*; *Huberfeld et al., 2007*; *Price et al., 2009*; *Hyde et al., 2011*; *Tyzio et al., 2014*). Similarly, intracellular $Cl^-$ is thought to be modulated during brain development so that GABAergic transmission contributes optimally to the construction of neural circuits (*Ben-Ari, 2002*). Given the importance of $Cl^-$ for brain function and dysfunction, the cellular mechanisms that control its transmembrane gradient and driving force are of considerable interest.

Plasmalemmal $Cl^-$ transporters, in particular cation-chloride cotransporters (CCCs), are understood to be the major mechanism by which neurons regulate the driving force for $Cl^-$ permeable anion channels (*Kaila et al., 2014*). Recently, it has been suggested that in fact impermeant anions

**eLife digest** Cells called neurons in the brain communicate by triggering or inhibiting electrical activity in other neurons. To inhibit electrical activity, a signal from one neuron usually triggers specific receptors on the second neuron to open, which allows particles called chloride ions to flow into or out of the neuron.

The force that moves chloride ions (the so-called 'chloride driving force') depends on two main factors. Firstly, chloride ions, like other particles, tend to move from an area where they are plentiful to areas where they are less abundant. Secondly, chloride ions are negatively charged and are therefore attracted to areas where the net charge (determined by the mix of positively and negatively charged particles) is more positive than their current position.

It was previously believed that a group of proteins known as CCCs, which transport chloride ions and positive ions together across the membranes surrounding cells, sets the chloride driving force. However, it has recently been suggested that negatively charged ions that are unable to cross the membrane (or 'impermeant anions' for short) may set the driving force instead by contributing to the net charge across the membrane. Düsterwald et al. used a computational model of the neuron to explore these two possibilities.

In the simulations, altering the activity of the CCCs led to big changes in the chloride driving force. Changing the levels of impermeant anions altered the volume of cells, but did not drive changes in the chloride driving force. This was because the flow of chloride ions across the membrane led to a compensatory change in the net charge across the membrane.

Düsterwald et al. then used an experimental technique called patch-clamping in mice and rats to confirm the model's predictions. Defects in controlling the chloride driving force in brain cells have been linked with epilepsy, stroke and other neurological diseases. Therefore, a better knowledge of these mechanisms may in future help to identify the best targets for drugs to treat such conditions.

DOI: https://doi.org/10.7554/eLife.39575.002

control local $[Cl^-]_i$ and driving force (*Glykys et al., 2014*), rather than the CCCs. The majority of intracellular anions are impermeant to the neuronal membrane; these include ribo- and deoxynucleotides, intracellular proteins and metabolites (*Burton, 1983*). Impermeant anions induce what is known as the Donnan (or Gibbs-Donnan) effect (*Hill, 1956*; *Sperelakis, 2012*) – an uneven distribution of impermeant molecules across the membrane which is osmotically unstable. Without active ion transport to counter this effect, neurons would swell and burst (*Kay, 2017*). Animal cells, including neurons, maintain cell volume in the presence of impermeant anions by using the $Na^+/K^+$-ATPase to pump $Na^+$ out of the cell and $K^+$ in, along with the passive movement of water and other ions (*Tosteson and Hoffman, 1960*; *Armstrong, 2003*; *Liang et al., 2007*; *Kay, 2017*). This pump-leak mechanism, whilst stabilizing cell volume, also establishes the negative resting membrane potential and transmembrane $Na^+$ and $K^+$ gradients, which serve as energy sources for the coupled transport of other molecules, including $Cl^-$ by CCCs. In the absence of active $Cl^-$ transport, $[Cl^-]_i$ is set by the membrane potential; i.e. the Nernst potential of $Cl^-$ ($E_{Cl} = \frac{RT}{F} \ln \frac{Cl_i}{Cl_o}$) equals the transmembrane potential ($V_m$).

The transmembrane $Cl^-$ gradient and the driving force (DF = $V_m - E_{Cl}$) for $Cl^-$ permeable ion channels are therefore the outcome of multiple, dynamically interacting mechanisms. This makes experimental investigation of the determinants of $Cl^-$ driving force difficult, particularly at a local level. Computational models based on established biophysical first principles are a productive means for exploring the roles of cellular mechanisms in generating local $Cl^-$ driving force. Here, we establish numerical and novel analytic solutions for an inclusive model of $Cl^-$ homeostasis to elucidate the determinants of the neuronal driving force for $Cl^-$.

We demonstrate that baseline $[Cl^-]_i$ is a product of the interaction of $Na^+/K^+$-ATPase activity, the mean charge of impermeant anions, ion conductances, CCCs and water permeability. Consistent with recent experimental reports (*Glykys et al., 2014*), and our own experimental validation using electroporation of anionic dextrans and optogenetic probing of $E_{GABA}$, we find that impermeant anions can contribute to setting $[Cl^-]_i$. However, we find that they can only affect the $Cl^-$ driving force by modifying active transport mechanisms, and then only negligibly. Impermeant anions therefore

do not appreciably modify synaptic signaling properties, contrary to the interpretation of recent experiments (*Glykys et al., 2014*). In contrast, we demonstrate using biophysical models and gramicidin perforated-patch clamp recordings that CCCs selectively regulate substantial changes in the Cl⁻ driving force. This is consistent with a meta-analysis of experimental data from the field, which shows a strong correlation between the activity of the specific CCC, KCC2, and Cl⁻ driving force. The ability of CCCs to specifically modulate Cl⁻ at a local level depends on the characteristics of Cl⁻ electrodiffusion in the structure concerned, demonstrated using multicompartment modeling. Together, our models provide a theoretical framework for understanding the interplay of chloride regulatory processes in neurons and interpreting experimental findings.

## Results

### A biophysical model based on the pump-leak mechanism demonstrates the importance of the sodium-potassium ATPase for setting transmembrane ion gradients including chloride

To compare the effects of impermeant anions and Cl⁻ cotransport on Cl⁻ homeostasis we first developed a single compartment model based on the pump-leak formulation (*Tosteson and Hoffman, 1960*; *Kay, 2017*) (*Figure 1A*). This model, defined by a set of differential equations, incorporated mathematical representations of the three major permeable ion species Cl⁻, K⁺, Na⁺ as well as impermeant ions ($X^z$) with mean charge z. Permeable ions could move across the cellular membrane via passive conductances according to each ion's respective electrochemical gradient. Further, active transport of Na⁺ and K⁺ by the Na⁺/K⁺-ATPase (with a 3:2 stoichiometry) and cotransport of Cl⁻ and K⁺ (1:1 stoichiometry) by the cation-chloride cotransporter KCC2 with a non-zero conductance $g_{KCC2}$ of 20 $\mu S/cm^2$ unless otherwise stated were included. Finally, our formulation accounted for the dynamics of cell volume (w), intracellular osmolarity ($\Pi$) and transmembrane voltage ($V_m$) (see Materials and methods). Importantly, regardless of initial starting concentrations of permeant or impermeant ions, cell volume or $V_m$, the model converged to stable fixed points without needing to include any means for 'sensing' ion concentration, volume or voltage. Initial permeable ion concentrations also did not influence the final cellular volume. For example, despite initiating the model with different starting concentrations of Cl⁻ (1, 15, 40 and 60 mM, respectively, *Figure 1B*), [Cl⁻]ᵢ always converged to the same stable concentration of 5.2 mM, a typical baseline [Cl⁻]ᵢ for adult neurons, and volume always converged to 2.0 pL, a typical volume for hippocampal neurons (*Ambros-Ingerson and Holmes, 2005*). The model is robust in the sense that its convergence to a stable steady state does not depend on a narrow set of parameters and initial values. Trying an alternative model of the Na⁺/K⁺-ATPase (*Hamada et al., 2003*) produced similar results (*Figure 1—figure supplement 1A-B*).

Consistent with previous results (*Xiao et al., 2002*; *Dierkes et al., 2006*; *Dijkstra et al., 2016*), 'turning off' the activity of the Na⁺/K⁺-ATPase in our model led to a progressive collapse of transmembrane ion gradients, progressive membrane depolarization and continuous and unstable cell swelling. Such effects could be reversed by reactivation of the pump (*Figure 1C*). The relative activity of the Na⁺/K⁺-ATPase sets the final stable values for ion concentrations (of both permeable and impermeant ions), $V_m$ and volume (*Figure 1C*). When we increased the activity of the Na⁺/K⁺-ATPase in our model, the final steady-state concentration for K⁺ increased, whilst Na⁺ and Cl⁻ dropped to levels that approximate those observed in mature neurons (*Figure 1D*). Note that although sufficient Na⁺/K⁺-ATPase activity is critical for steady state ionic gradients including that of Cl⁻, these are relatively stable near the default pump rate (*Figure 1—figure supplement 2*). At the same time, the final, stable-state membrane potential and cell volume also decreased. Interestingly, as has been observed previously (*Fraser and Huang, 2004*), beyond a certain level, further increases in Na⁺/K⁺-ATPase activity have negligible effects on cell volume and transmembrane voltage. For subsequent analysis, we chose a 'default' effective pump rate for the Na⁺/K⁺-ATPase of approximately $1.0 \times 10^{-2}$ C/(dm².s), that is a pump rate constant of $10^{-1}$ C/(dm².s) (*equation (2)*), and mean intracellular impermeant anion charge (z) of −0.85 as extrapolated from reasonable cellular ionic concentrations and osmolarity (*Lodish et al., 2009*; *Raimondo et al., 2015*). This resulted in steady-state ion concentrations and membrane potentials that approximate those experimentally observed in mature neurons: Cl⁻ 5 mM; K⁺ 123 mM; Na⁺ 14 mM; $X^z$ 155 mM; and a $V_m$ of −72.6 mV (*Jiang and Haddad,*

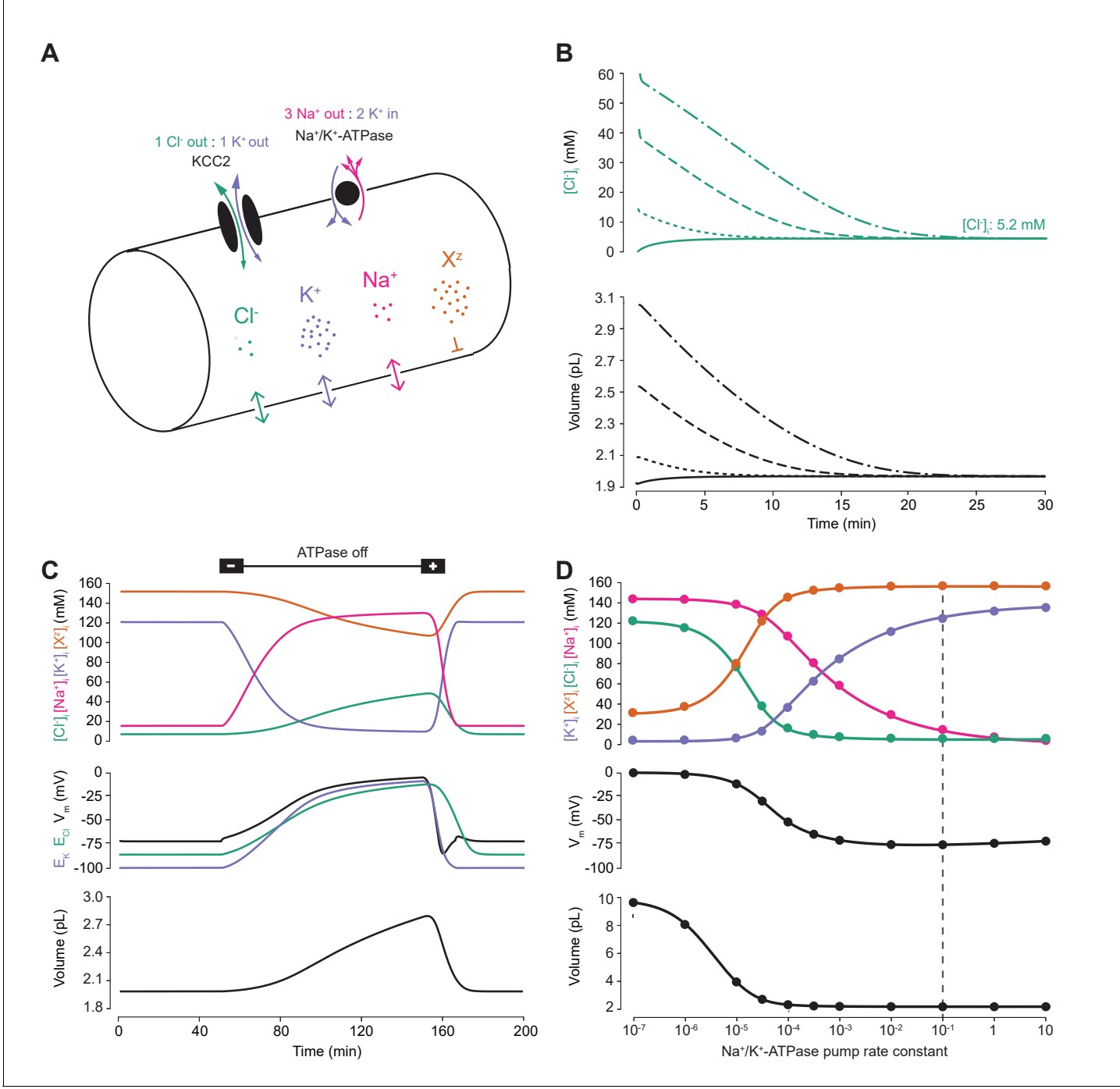

**Figure 1.** A biophysical model of ion dynamics based on the pump-leak mechanism demonstrates the importance of the sodium-potassium ATPase for setting transmembrane ion gradients including chloride. (A) A single-cell compartment was modeled as a cylinder with volume changes equivalent to changes in cylindrical radius. Dynamics of membrane permeable potassium (purple, $K^+$), sodium (pink, $Na^+$) and chloride (green, $Cl^-$) ions were included. Impermeant anions ($X^z$, orange) had a mean intracellular charge z of -0.85. The KCC2 transporter moved $Cl^-$ and $K^+$ in equal parts according to the transmembrane gradient for the two ions. The $Na^+$/ $K^+$ ATPase transported 3 $Na^+$ ions out for 2 $K^+$ ions moved into the cell. (B) Regardless of intracellular starting concentrations of the permeable ions, the model converged to identical steady state values for all parameters without needing to include any means for 'sensing' ion concentration, volume or voltage. We show the result for $Cl^-$ as a time series of $[Cl^-]_i$ (top panel) and volume (bottom panel). (C) The ATPase plays a key role in maintaining steady state ion concentration, membrane voltage ($V_m$) and volume. Switching off the ATPase results in a continuous increase in cell volume (bottom panel), membrane depolarization (middle panel) and ion concentration dysregulation (top panel with colours per ion as in 'A'). All cellular parameters recovered when the ATPase was reactivated. (D) The model's analytic solution showed exact correspondence with steady state values generated by numerical, time series runs (dots) for varying ATPase pump rates. Steady state values for

*Figure 1 continued on next page*

*Figure 1 continued*

the concentrations of the ions with colours as in 'A' (top panel); $V_m$ (middle panel) and volume (bottom panel). The dashed line indicates the default ATPase pump rate used in all simulations unless specified otherwise. The result in 'B' was replicated with Hamada et al.'s experimentally validated model of the ATPase in *Figure 1—figure supplement 1B*, and the models' respective pump fluxes are compared in *Figure 1—figure supplement 1A*. Note that although sufficient $Na^+/K^+$-ATPase activity is critical for steady state ionic gradients these variables are relatively stable near the default pump rate (dashed line), *Figure 1—figure supplement 2*.

DOI: https://doi.org/10.7554/eLife.39575.003

The following figure supplements are available for figure 1:

**Figure supplement 1.** Different models of the sodium-potassium ATPase have distinct kinetic properties but produce similar qualitative effects.

DOI: https://doi.org/10.7554/eLife.39575.004

**Figure supplement 2.** Changes in $Na^+/K^+$-ATPase activity near the model's default pump rate produce minimal changes in $E_{Cl}$ and $V_m$.

DOI: https://doi.org/10.7554/eLife.39575.005

*1991*; *Diarra et al., 2001*; *Tyzio et al., 2008*). We were able to corroborate the numerical solutions for final steady-state values by developing a parametric-analytic solution (*Supplementary file 1*). We observed exact correspondence between the numerical and analytic solutions within our model (*Figure 1D*). In subsequent analyses, this novel analytic solution allowed us to explore rapidly a large parameter space to determine how various cellular attributes might affect $Cl^-$ homeostasis.

## Membrane chloride conductance affects steady-state intracellular chloride concentration only in the presence of cation-chloride cotransport

Using the analytic solution, we investigated how changes in baseline ion conductance for the major ions in our model ($g_K$, $g_{Na}$ and $g_{Cl}$) affected $Cl^-$ homeostasis. We calculated the steady-state values for the $Cl^-$ reversal potential ($E_{Cl}$) and $K^+$ reversal potential ($E_K$), resting membrane potential ($V_m$) and volume (w) whilst independently manipulating the conductance for each ion (*Figure 2*). Increasing the baseline $K^+$ conductance ($g_K$) resulted in $E_{Cl}$, $E_K$ and $V_m$ converging to similar steady-state values (*Figure 2A*) without significantly affecting cell volume. We were also able to replicate the classic dependence of membrane potential on $\log([K^+]_o)$ (*Figure 2—figure supplement 1*). In contrast, increasing the baseline $Na^+$ conductance ($g_{Na}$) beyond 20 $\mu S/cm^2$ resulted in a steady increase of $E_{Cl}$, $V_m$ and volume with a minimal increase of $E_K$ (*Figure 2B*). $E_K$ is maintained in the face of increased passive $K^+$ efflux accompanying membrane depolarization due to increased active influx of $K^+$ by the $Na^+$-dependent ATPase which increases its effective pump rate due to increased intracellular $Na^+$ concentration with larger $g_{Na}$ (*Figure 2—figure supplement 2*).

The effect of manipulating $Cl^-$ conductance ($g_{Cl}$) depended on the activity of concurrent cation-chloride cotransport by KCC2 (*Figure 2C*). In the presence of active KCC2 at very low values of $g_{Cl}$, the steady state $[Cl^-]_i$ is such that $E_{Cl}$ approaches $E_K$. This follows because in the absence of alternative $Cl^-$ fluxes, KCC2 utilizes the transmembrane $K^+$ gradient to transport $Cl^-$ until $E_{Cl}$ equals $E_K$. With increasing $g_{Cl}$ however, $E_{Cl}$ increases, moving away from $E_K$ toward $V_m$, and at very high $Cl^-$ conductances $E_{Cl}$ and $V_m$ approached similar values in our model. Without the activity of KCC2, any nonzero $g_{Cl}$ had no effect on steady state $E_{Cl}$, $E_K$, $V_m$ or volume (*Figure 2D*). In this instance $E_{Cl}$ always equals $V_m$ as the movement of $Cl^-$ across the membrane is purely passive. Without the activity of KCC2, there can be no driving force for $Cl^-$ flux at steady state ($V_m$-$E_{Cl}$ = 0). Our model therefore behaved in a manner consistent with established theoretical predictions (*Kaila et al., 2014*).

## Cation-chloride cotransport sets the chloride reversal and driving force for transmembrane chloride flux

Next, we used our single-cell unified model to explore how the activity of cation-chloride cotransport affects $Cl^-$ homeostasis. In our model, the activity of KCC2 is set by the conductance of KCC2 ($g_{KCC2}$). Using the numerical formulation with the default values described in *Figures 1* and *2*, we steadily increased $g_{KCC2}$ from 20 $\mu S/cm^2$ to 370 $\mu S/cm^2$ and tracked changes to $E_{Cl}$, $E_K$, $V_m$ and volume. Increasing KCC2 activity over time caused a steady decrease in $[Cl^-]_i$ reflected by a hyperpolarization of $E_{Cl}$ (*Figure 3A*). $V_m$ decreased only modestly, resulting in an increase in the driving force for $Cl^-$ flux that tracks the increase in $g_{KCC2}$. This effect saturates as $E_K$ constitutes a lower bound on $E_{Cl}$. Importantly, increases in $g_{KCC2}$ resulted in persistent changes to $E_{Cl}$ and the driving force for $Cl^-$.

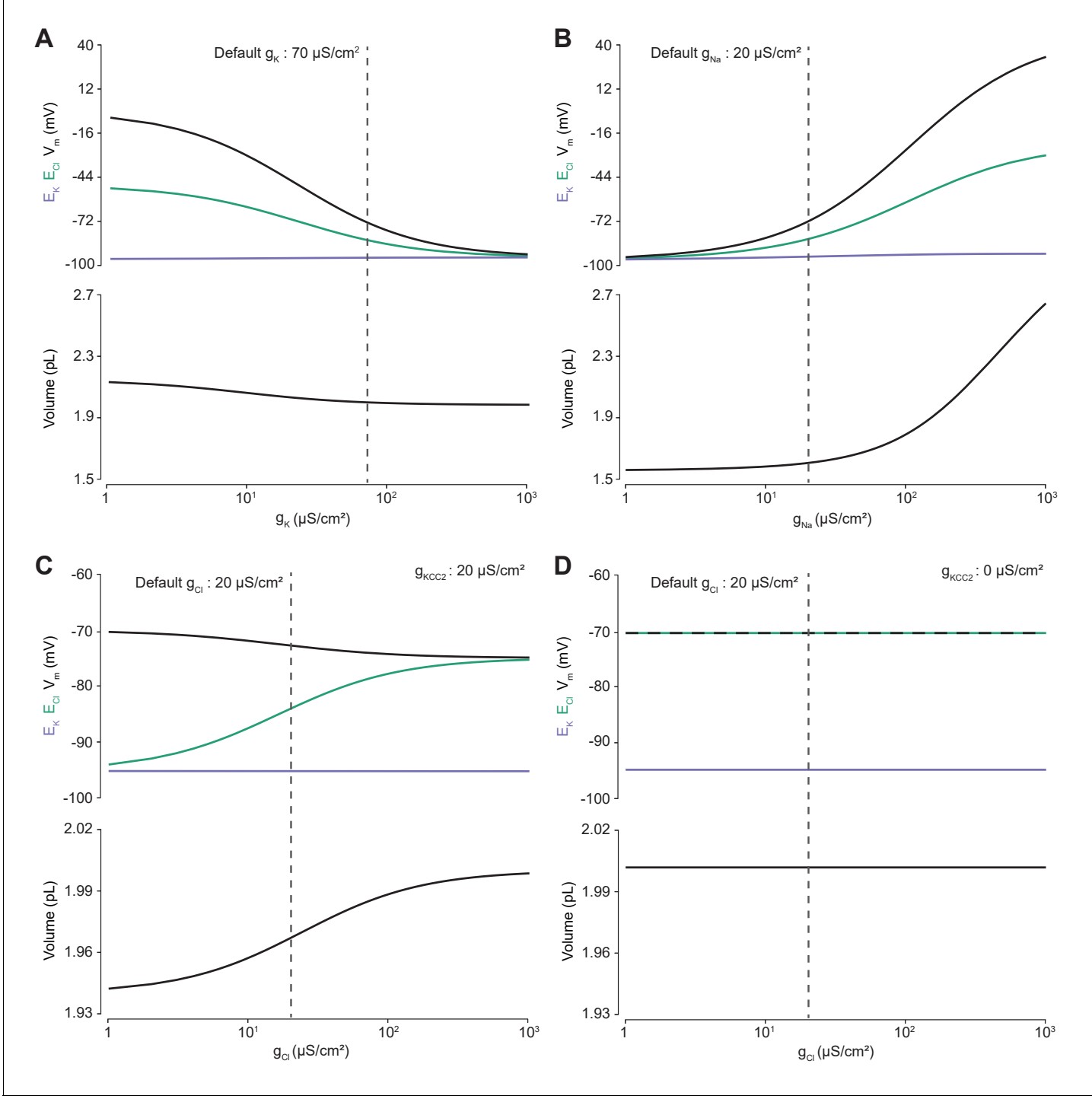

**Figure 2.** Membrane conductances affect steady-state intracellular chloride concentration only in the presence of cation-chloride cotransport. Steady state values for different ionic conductance were calculated using the model's analytic solution. (A) Steady state $E_{Cl}$ (green), $E_K$ (purple), $V_m$ (black) and volume w (bottom panel) were calculated at different $K^+$ conductances ($g_K$). Increasing $g_K$ resulted in a convergence of steady state $E_{Cl}$ and $V_m$. (B) Increasing $Na^+$ conductance ($g_{Na}$) resulted in a progressive increase in steady state $E_{Cl}$, $V_m$ and volume with a negligible increase in $E_K$. *Figure 2—figure supplement 2* demonstrates how increased $K^+$ flux through the $Na^+/K^+$ ATPase matches passive $K^+$ efflux to maintain $E_K$. (C) In the presence of active cation-chloride cotransport ($g_{KCC2}$ = 20 μS/cm²), increasing $Cl^-$ conductance shifted steady state $E_{Cl}$ from $E_K$ toward $V_m$. (D) In the absence of KCC2 activity, $g_{Cl}$ had no effect on steady state parameters. $E_{Cl}$ equals $V_m$ in all instances. Dashed lines indicate the default values for $g_K$, $g_{Na}$ and $g_{Cl}$. In *Figure 2—figure supplement 1* the classic dependence of the membrane potential on log($[K^+]_o$) is shown.

DOI: https://doi.org/10.7554/eLife.39575.006

*Figure 2 continued on next page*

*Figure 2 continued*

The following figure supplements are available for figure 2:

**Figure supplement 1.** The classic dependence of membrane potential (mV) on the logarithm of extracellular potassium is reproduced using our model.
DOI: https://doi.org/10.7554/eLife.39575.007

**Figure supplement 2.** $K^+$ flux is balanced by active and passive fluxes, dependent on $Na^+$ steady states.
DOI: https://doi.org/10.7554/eLife.39575.008

Employing alternate models for KCC2 (*Fraser and Huang, 2004*; *Lewin et al., 2012*; *Raimondo et al., 2012*) and the $Na^+/K^+$-ATPase (*Hamada et al., 2003*) did not change this result when compensation for parameterization was given, although different KCC2 models result in different kinetic rates for $Cl^-$ and $K^+$ transport (*Figure 3—figure supplements 1* and *2*). Using the analytic solution to our model, we calculated how KCC2 activity affects steady state values of $E_{Cl}$, $E_K$, $V_m$, volume and $Cl^-$ driving force (*Figure 3B*). In confirmation of our findings in *Figure 2D*, with no KCC2 activity ($g_{KCC2} = 0$), $E_{Cl}$ equaled $V_m$ and the $Cl^-$ driving force was zero. As we increased $g_{KCC2}$, steady state $E_{Cl}$ pulled away from $V_m$ and approached $E_K$. This resulted in an increase in $Cl^-$ driving force ($V_m$-$E_{Cl}$) with steady state values of 11.3 mV at our chosen default value of $g_{KCC2}$. The results obtained with our model are therefore fully consistent with the view that CCCs, in this case KCC2, establish the driving force for $Cl^-$.

To test this theoretical finding, we performed gramicidin perforated patch-clamp recordings from CA3 hippocampal neurons in rat organotypic brain slices whilst activating $Cl^-$ permeable $GABA_A$ receptors with muscimol (10 µM), in order to measure the $GABA_A R$ driving force, which approximates $Cl^-$ driving force (*Figure 3C*). We then tested our model predications by applying the CCC blocker, furosemide (1 mM) (*Figure 3D,E*). We noted that after furosemide was introduced the $E_{GABA}$ became significantly more depolarized (baseline: $-78.8 \pm 2.8$ mV vs furosemide: $-71.6 \pm 3.0$ mV, $n = 10$, p=0.01, *paired t-test*), whilst there was no significant difference in $V_m$ ($-69.9 \pm 1.8$ mV vs $-71.5 \pm 2.6$ mV, $n = 10$, p=0.36, *paired t-test*) (*Figure 3E*). This reflects a significant change in the $GABA_A R$ driving force ($8.9 \pm 3.4$ mV vs $0.1 \pm 4.6$ mV, $n = 10$, p=0.04, *paired t-test*). Values were stable prior to baseline recordings with $E_{GABA}$ 5 min prior to furosemide application at $-78.4 \pm 8.1$ mV, $V_m$ $-70.1 \pm 5.6$ mV and DF at $9.1 \pm 3.2$ (see *Figure 3E*). We then noted that this change in driving force persisted for at least 15 min post-application of furosemide (*Figure 3E*). These results are consistent with our model predictions and demonstrate how the application of a CCC blocker reduces the $GABA_A R$ driving force (and hence $Cl^-$ driving force) by selectively depolarizing $E_{GABA}$ with negligible effects on $V_m$.

In addition, we sought experimental data from the literature to determine whether changes in KCC2 activity correlate with alterations to steady-state $[Cl^-]_i$. We focused on changes in KCC2 expression level, as this is likely to be a strong predictor of changes in KCC2 activity. Indeed, in a meta-analysis of seven studies and eight experiments from our review of 26 studies, weighted for methodological biases and data quality, we observed a significant correlation ($R^2 = 0.796$, p<0.001) between the change in KCC2 expression and $Cl^-$ driving force (*Figure 3F*). Absolute changes in $V_m$ were less than 2 mV in all but one study, meaning that the change in driving force could be ascribed to significant shifts in $E_{GABA}$ ($R^2 = 0.045$, p<0.001). The outlier data point (showing a 8.45 mV change) was from a study into the effects of acute stress, where other factors could have transiently influenced $V_m$ (*MacKenzie and Maguire, 2015*). The meta-analysis supports the prediction that cation-chloride cotransport by KCC2 is an important determinant of $[Cl^-]_i$ ($R^2 = 0.83$, p<0.001, nine studies) and driving force (see *Supplementary file 2* Table S2-1 for raw data, and the scoring table for weighting in Table S2-2).

## Altering the concentration of intracellular or extracellular impermeant anions, without changing the mean charge of impermeant anions, does not affect the steady state gradient or driving force for chloride

To determine the effect of impermeant anions on $Cl^-$ homeostasis, we first explored whether adjusting the concentration of impermeant anions ($[X]_i$), while maintaining a constant mean impermeant ion charge (z), had any impact on $E_{Cl}$, $E_K$, $V_m$ or volume. The mean charge (z) is the mean of the charge of all the different species of impermeant molecules in the cell, including uncharged ones,

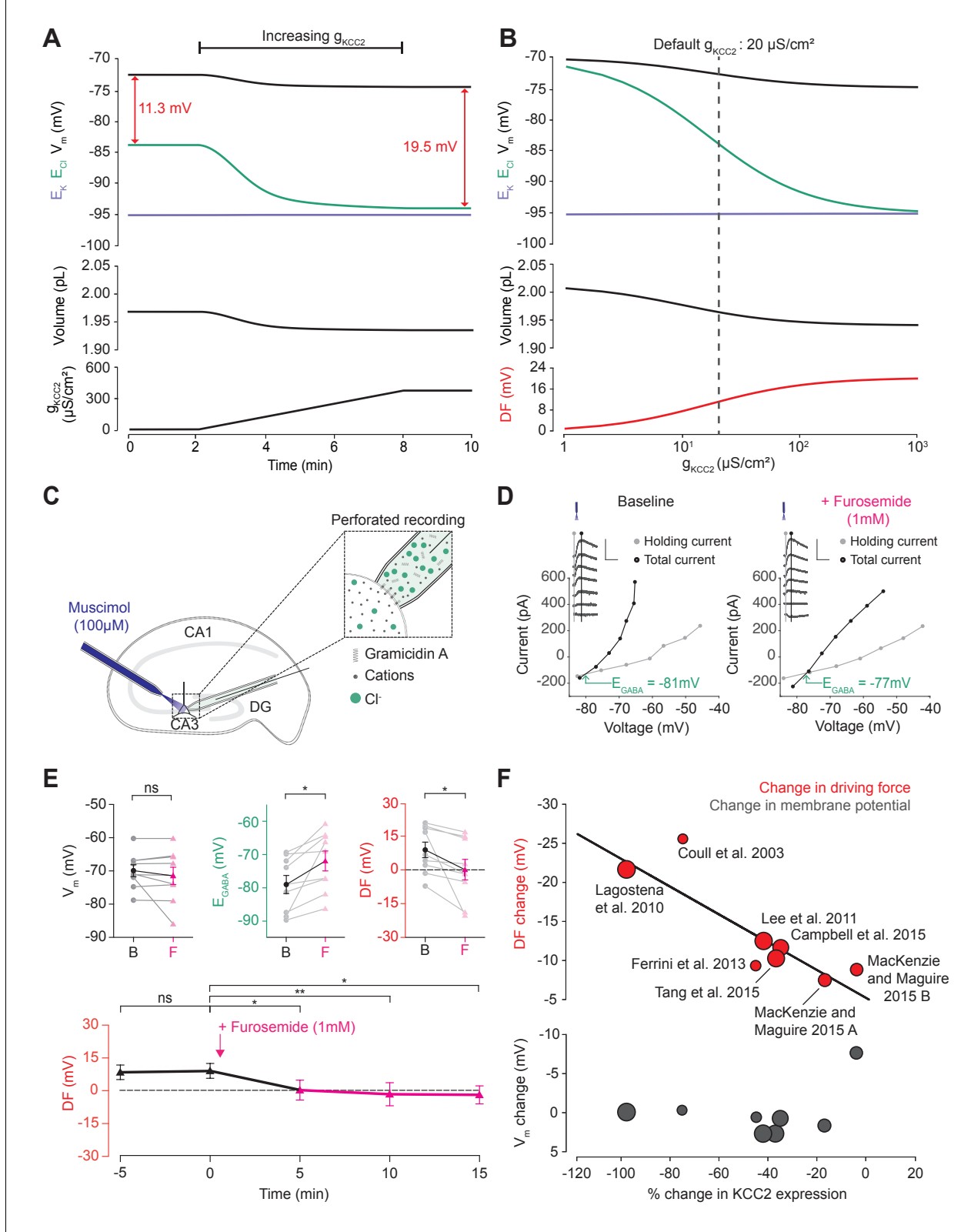

**Figure 3.** Cation-chloride cotransport sets the chloride reversal and driving force for transmembrane chloride flux. (**A**) Increasing KCC2 activity in our model by increasing $g_{KCC2}$ from 20 $\mu$S/cm$^2$ to 370 $\mu$S /cm$^2$ resulted in a persistent decrease in $E_{Cl}$ (green), a minimal decrease in $E_K$ (purple), $V_m$ (black) and volume w (bottom panel). This resulted in a permanent increase in the DF for Cl$^-$ via a change in $E_{Cl}$ from $-83.9$ mV to $-93.2$ mV (red). (**B**) The steady state values for $E_{Cl}$, $E_K$, $V_m$ (top panel), volume (middle panel) and Cl$^-$ driving force (DF, bottom panel, red) at different KCC2 conductances.

*Figure 3 continued on next page*

*Figure 3 continued*

Increasing KCC2 activity resulted in a decrease in steady state $E_{Cl}$ and an increase in DF. $E_K$ represented a lower bound on $E_{Cl}$ at high KCC2 conductances. Similar results were noted for other kinetic models of KCC2 (*Figure 3—figure supplements 1* and *2*). (C) Schematic showing experimental setup. Gramicidin perforated patch-clamp recordings were performed on CA3 pyramidal cells from rat hippocampal organotypic brain slices. (D) Insets depict GABA$_A$R currents elicited by somatic application (20 ms) of muscimol (10 µM) at different voltages. Calibration: 500 ms, 500 pA. Holding current (reflecting membrane current) and total current (reflecting membrane current plus the muscimol-evoked current) were measured at the points indicated by the vertical grey and back lines, respectively. Current-voltage (IV) plots were drawn to calculate changes in $V_m$, $E_{GABA}$ and DF before (left) and after (right) furosemide (pink) was applied. Voltages were corrected for series resistance error. (E) Top, population data showing significant changes in $E_{GABA}$ and DF but not $V_m$ five minutes after furosemide application. Bottom, changes in DF over time show a significant decrease from baseline once furosemide was introduced. (F) Meta-analysis of experimental studies demonstrates a correlation between KCC2 activity (% change) and Cl$^-$ DF (mV, top plot, red) but not membrane potential (mV, bottom plot, grey), confirming the role of KCC2 for establishing the neuronal Cl$^-$ gradient in adult tissue. The data and scoring system used to generate the regression can be found in *Supplementary file 2* (Tables S2-1 and S2-2 'ns', non-significant; *p<0.05; **p<0.01. The data for 'C', 'D' and 'E' is provided in *Figure 3—source data 1*.

DOI: https://doi.org/10.7554/eLife.39575.009

The following source data and figure supplements are available for figure 3:

**Source data 1.** Source data for *Figure 3C–E*.

DOI: https://doi.org/10.7554/eLife.39575.012

**Figure supplement 1.** Models of KCC2 have different kinetic properties.

DOI: https://doi.org/10.7554/eLife.39575.010

**Figure supplement 2.** KCC2 and ATPase models with different kinetics have similar properties when KCC2 is upregulated.

DOI: https://doi.org/10.7554/eLife.39575.011

where charge is the difference between the number of protons and electrons of a molecule. Impermeant anions are more abundant than impermeant cations, and so in this manuscript we often refer to the group as impermeant anions rather than impermeant ions or impermeant molecules. For example, were there $\alpha$ impermeant molecules of charge $-1$ and $\beta$ impermeant molecules of charge 0, then z would be $\frac{-\alpha}{\alpha+\beta}$. We initiated the full single-compartment model with different starting concentrations of impermeant anions all with the same mean charge, z = $-0.85$, and observed that regardless of the initial concentration of impermeant anions, over a period of minutes, the cell adjusted its volume to give an identical steady-state impermeant anion concentration (*Figure 4A*, $[A]_i$ = 155 mM). Analytically, it can be shown that the number of moles of X determines completely the volume of the compartment, while the permeant ions alone cannot be used to predict steady state volume (*Kay, 2017*). Similarly, all initial impermeant anion concentrations resulted in identical steady state values of $E_{Cl}$ ($-83.8$ mV), $E_K$ ($-95.1$ mV) and $V_m$ ($-72.6$ mV) (*Figure 4B*). This shows that simply adjusting the amount of impermeant anions within a cell has no persistent effect on $[Cl^-]_i$.

We then tested the effect of dynamically adding impermeant anions with the default mean charge either intracellularly (*Figure 4C*) or extracellularly (*Figure 4D*). While impermeant anions are being added to the cell, the membrane potential hyperpolarizes and $E_{Cl}$ decreases. However, following the cessation of impermeant anion influx, $E_{Cl}$, $E_K$, $V_m$ and $[X]_i$ return to steady state values due to compensatory changes to cell volume (*Figure 4C*). There are transient transmembrane fluxes of all ions while anions are added into the cell, and in particular the inward flux of the cations Na$^+$ and K$^+$, such that the sum $[X]_i + [Cl^-]_i$ is not necessarily kept constant during impermeant anion addition (*Figure 4—figure supplement 1*). Impermeant ions (with the default mean charge) were added to the extracellular space, which is effectively an infinite bath in the model, while proportional decreases in $[Cl^-]_o$ were applied to correct for the changes to charge and osmotic balance. Additions in the extracellular space, similarly, resulted in a temporary depolarization of $E_{Cl}$ and $V_m$, but no persistent shift in these parameters (*Figure 4D*). The addition of extracellular impermeant anions did however result in a small compensatory decrease in cell volume secondary to the large shifts in $[Cl^-]_i$ required to maintain the proportion of $[Cl^-]_i$ to $[Cl^-]_o$ according to the Nernst potential. In summary, there is no lasting effect on the reversal potential or driving force for Cl$^-$ if only the concentration of a neuron's intracellular or extracellular impermeant anions is altered. This is because concentration changes alone modulate only osmoneutrality, whereas changes to intracellular charge balance affect electroneutrality and therefore the membrane and ionic potentials, which we tested next.

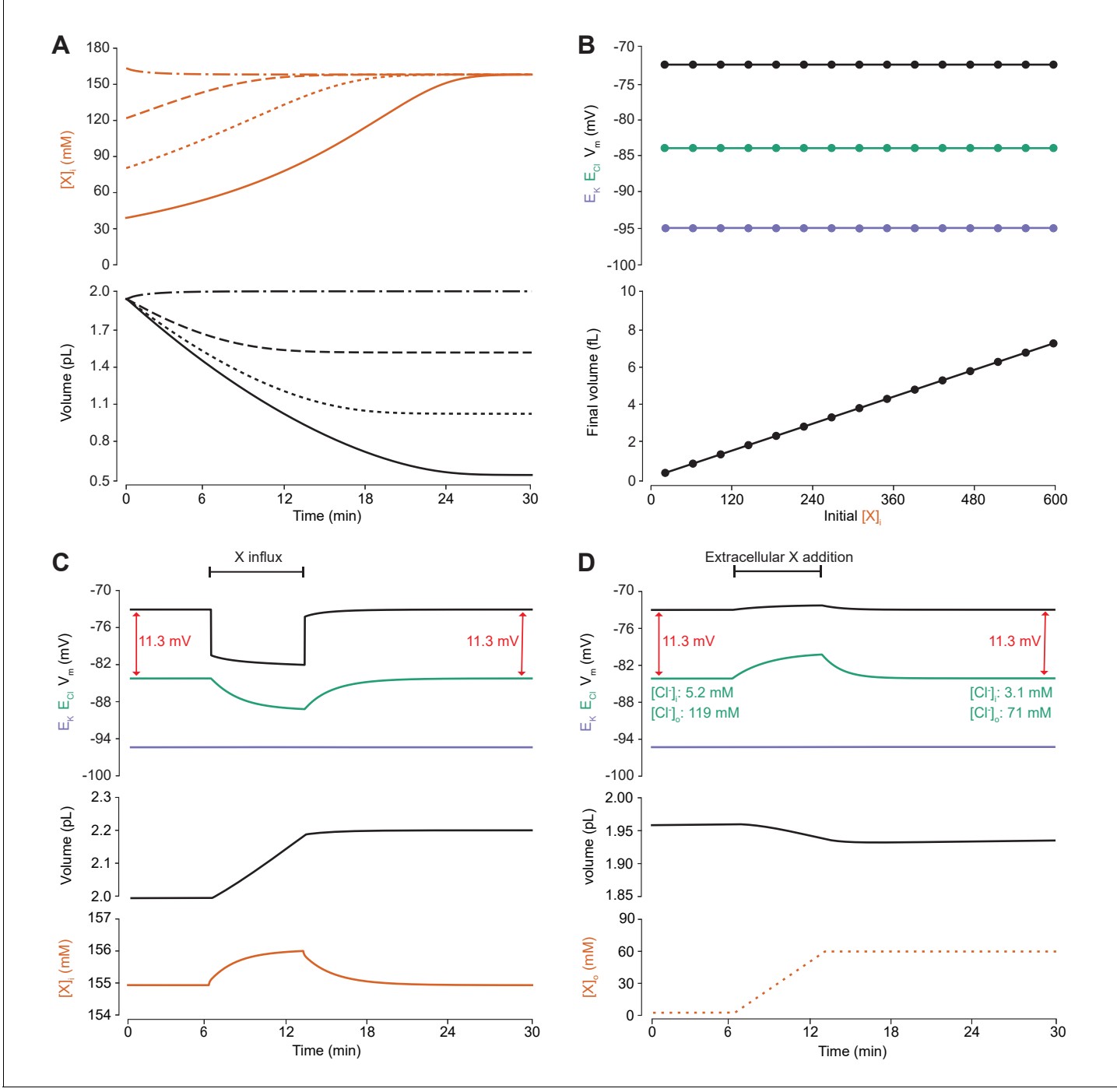

**Figure 4.** Adding intracellular or extracellular impermeant anions, without changing the mean charge of impermeant anions, does not affect the steady state gradient or driving force for chloride. (**A**) Initiating the model with different starting concentrations of intracellular impermeant anions ($[X]_i$) with the same mean charge $z = -0.85$ (orange, top panel), led to compensatory volume changes (bottom panel) which resulted in identical steady state concentrations. (**B**) Steady state $E_{Cl}$ (green), $E_K$ (purple) and $V_m$ (black) were identical regardless of initial $[X]_i$. Final volume, however, was a linear function of initial $[X]_i$ (bottom panel). (**C**) Addition of impermeant anions of the mean charge ($z = -0.85$) caused transients shifts in $E_{Cl}$ (green, top panel), $E_K$ (purple), $V_m$ (black) as well as $[X]_i$ (orange, bottom panel) for the duration of the influx, and sustained increases in volume (black, middle panel). No persistent changes in $E_{Cl}$, $E_K$ or $V_m$ were observed. The full time-dependent ionic and water fluxes for the experiment are shown in *Figure 4—figure supplement 1*, which shows that the inward flux of impermeant anions causes fluxes of all other ions. (**D**) Similarly, the addition of extracellular impermeant anions in an osmoneutral manner causes transient shifts in the permeable ion gradients (top panel, colours as in 'C'), and sustained changes in cellular volume (black, middle panel) as well as the extracellular X and extracellular and intracellular Cl⁻ concentrations.
DOI: https://doi.org/10.7554/eLife.39575.013

*Figure 4 continued on next page*

*Figure 4 continued*

The following figure supplement is available for figure 4:

**Figure supplement 1.** Flux of all ions for the manipulation in *Figure 4c*.

DOI: https://doi.org/10.7554/eLife.39575.014

## Changing the mean charge of impermeant anions can drive substantial shifts in the reversal potential for chloride, but has negligible effects on chloride driving force

We next sought to determine how changes in the mean charge of the impermeant ions (z) might influence the driving force for $Cl^-$. Such changes in z could be associated with various cellular processes, including post-translational modifications of proteins that decrease their charge without changing the absolute number of protein molecules. To investigate this parameter, we modified the mean charge (z) of intracellular impermeant anions from −0.85 to −1 whilst measuring accompanying changes in $E_{Cl}$, $E_K$, $V_m$ and cell (*Figure 5A*). We found that this shift to a more negative z resulted in both a transient and persistent decrease in $E_{Cl}$, $E_K$ and V. Importantly, the shifts in $E_{Cl}$ were accompanied by broadly matching shifts in $E_K$ and V, which resulted in a small change in the driving force for $Cl^-$ of <0.2 mV. Both numerical and analytic calculation of steady state values for $E_{Cl}$, $E_K$ and $V_m$ in our model showed that changing the mean charge of impermeant anions, while substantially affecting $E_{Cl}$, had very small effects on the driving force for $Cl^-$ (*Figure 5B*). By shifting z within reasonable ranges for mammalian neurons (*Lodish et al., 2009*; *Raimondo et al., 2015*), and assuming osmo- and electro-neutrality, only shifts of <1 mV could be generated. In addition, although the absolute number of impermeant anions (moles) remained constant throughout the process of modifying z, cell volume shifted, and as a consequence modest alterations to the concentration of impermeant anions occurred as well.

Next, instead of adjusting the charge of some of the intracellular impermeant anions as described above, we directly added new impermeant anions to the cell, which had a more negative charge than the previous mean charge (*Figure 5C and D*). This had the effect of both increasing the absolute quantity of impermeant anions and adjusting the mean charge of impermeant anions. The 'addition' of impermeant anions in this way models the de novo synthesis of impermeant anion species, or their active transport into the cell. This process also resulted in both transient and persistent changes to $E_{Cl}$, $E_K$ and V, which was dependent on the extent that z was altered. Again, whilst the large additions of impermeant anions could substantially alter the $Cl^-$ reversal potential, this had a negligible effect on the driving force for $Cl^-$ due to matching shifts in $V_m$. Driving shifts in $E_{Cl}$ in this manner also resulted in changes to cell volume (*Figure 5—figure supplement 1*).

To experimentally test our biophysical modeling predictions, we used photo-activation of ChR2 expressing GABAergic interneurons and whole-cell patch clamp recordings of mouse organotypic CA3 hippocampal pyramidal neurons to measure $V_m$, $E_{GABA}$, and GABA driving force (which approximates $Cl^-$ driving force) before and after addition of impermeant anions (*Figure 5D*). To add impermeant anions to the recorded cell, we used single-cell electroporation of fluorescently tagged anionic dextrans (Alexa Flour 488, see Materials and methods). Successful addition of impermeant anions could be confirmed visually by observing strong and stable fluorescence of the anionic dextran restricted to the recorded cell (*Figure 5D,E*). To drive impermeant anions into the cell, negative voltage steps (20 ms, 0.5–1 V) were applied to the electroporation pipette necessarily resulting in direct membrane depolarization, which recovered over a period of 1–5 min (*Figure 5D*). Once this acute perturbation had settled, as predicted by our model, changing the mean charge of impermeant anions, following the addition of highly negatively charged dextrans to the cell, resulted in a stable, mean negative shift in $V_m$ from a baseline of −67.0 ± 4.0 mV to −74.0 ± 3.1 mV (n = 6, p=0.03, *Wilcoxon test*, *Figure 5E,F*). Again in line with our predictions, addition of impermeant anions also resulted in a significant reduction of resting $E_{GABA}$ (a proxy for $E_{Cl}$) from baseline values of −72.5 ± 2.0 mV to −77.3 ± 1.4 mV (p=0.03, *Wilcoxon test*). Importantly, however, similar shifts in $V_m$ and $E_{GABA}$ resulted in an undetectable shift in GABA and hence $Cl^-$ driving force (5.5 ± 4.4 mV vs 3.3 ± 3.5 mV, n = 6, p=0.22, *Wilcoxon test*, *Figure 5E,F*).

Our single-compartment model of $Cl^-$ homeostasis, in conjunction with experimental validation, demonstrates that whilst the adjustment of mean impermeant anion charge can significantly affect

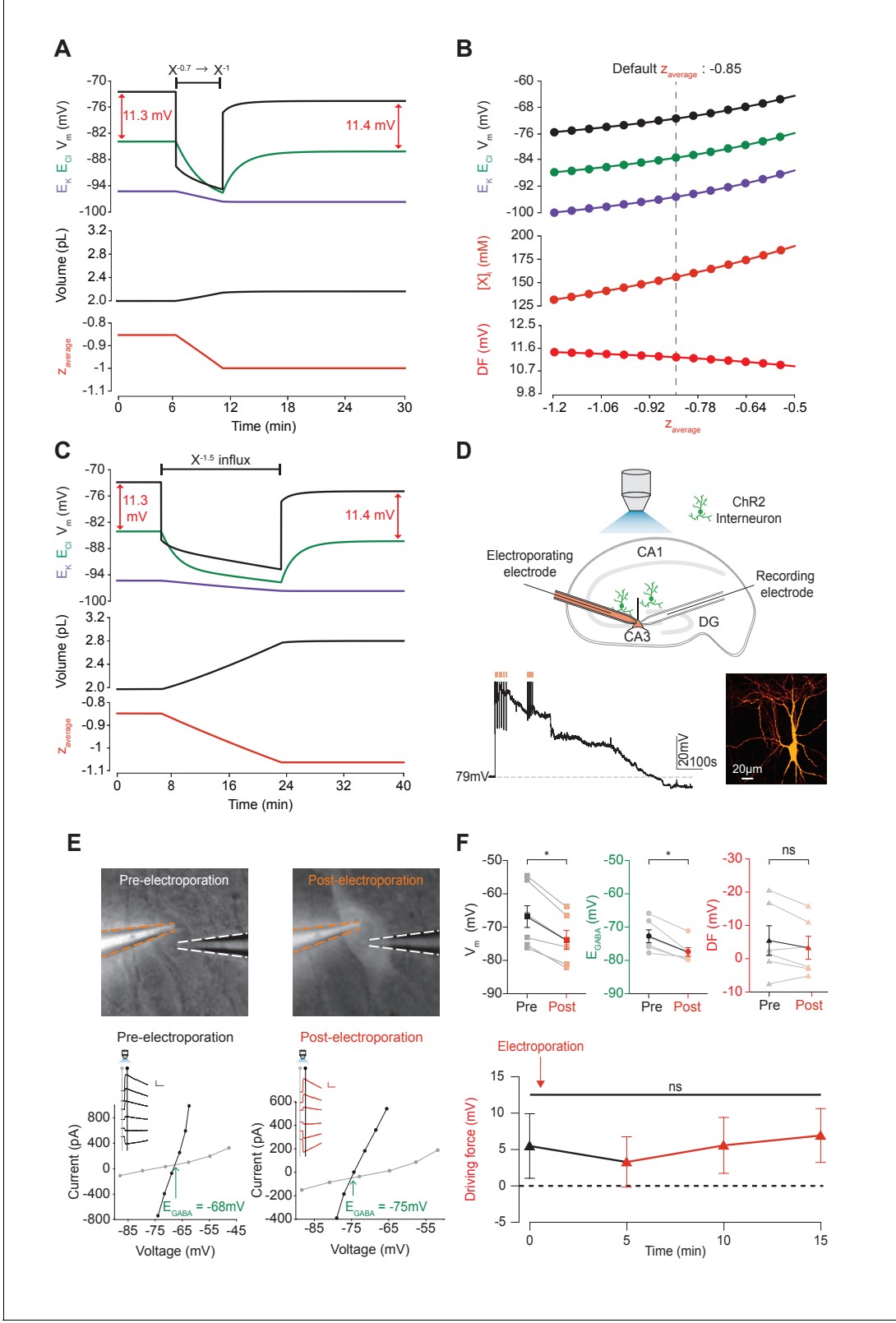

**Figure 5.** Adjusting the mean charge of impermeant anions shifts the chloride reversal potential with negligible effects on the driving force for chloride. (A) Decreasing the mean charge of impermeant anions from −0.85 (the default value) to −1 (orange, bottom panel), without changing the absolute number of intracellular impermeant anions caused a persistent decrease in $E_{Cl}$ (green, top panel), $E_K$ (purple) and $V_m$ with moderate increases in volume (middle panel) in our default single compartment model. Negligible changes in Cl⁻ driving force (ΔDF = 0.16 mV, red) were observed. (B)

*Figure 5 continued on next page*

*Figure 5 continued*

Analytic solution (solid lines) for different impermeant anion mean charge (z) exactly matches the steady state values from numerical, time series runs (dots) based on adjusting z as in 'A'. Steady state $E_{Cl}$, $E_K$, $V_m$ (top panel) and $[X]_i$ (middle) increased with increasing z, while changes in z resulted in very small changes in $Cl^-$ DF (bottom, red). The vertical dashed line indicates the values at the chosen default z of −0.85. (C) Influx of a species of impermeant anions with a charge of −1.5, that decreased the mean charge z (bottom panel) from −0.85 to −1 and increased the number of impermeant anions also caused persistent decreases of $E_{Cl}$, $E_K$ and $V_m$ as in 'A', but with larger increases in cell volume (middle panel). Again, very small persistent changes in $Cl^-$ DF were observed. The volume changes for different flux amounts and charges are illustrated in *Figure 5—figure supplement 1*. (D) Top, schematic of the experimental setup where whole-cell recordings were made from CA3 pyramidal cells in mouse organotypic brain slices. Impermeant anions (orange) were delivered via electroporation of the negatively charged fluorescent dextran Alexa Flour 488 via a pipette positioned near the soma of the recorded cell. $GABA_AR$ currents were elicited via photo-activation (100 ms, 470 nm LED via objective) of ChR2-expressing GAD2+ interneurons (green cells) in the presence of 5 µM CGP-35348 to block $GABA_BRs$. Lower trace, current clamp recording showing $V_m$ changes during electroporation of anionic dextran. Confocal image demonstrating cell-localized fluorescence of the anionic dextran electroporated in 'E'. (E) Top, widefield images with electroporation pipette (orange dashed lines) and the recording pipette (white dashed lines). Note increased fluorescence in the soma after electroporation. Below, insets show $GABA_AR$ currents evoked by photo-activation of GAD2+ interneurons at different holding potentials. Calibration: 1 s, 100 pA. Holding current (reflecting membrane current) and total current (reflecting membrane current plus the $GABA_AR$ current) were measured at the points indicated by the vertical grey and black lines, respectively. IV plots were used to calculate $V_m$, $E_{GABA}$ and DF before (left) and after (right) electroporation. (F) Top, population data showing significant decreases in mean $V_m$ and $E_{GABA}$ but not DF five minutes after electroporation. Below: changes in DF over time. Point at which electroporation occurred marked with orange arrow. 'ns', non-significant; *p<0.05. The data for 'D', 'E' and 'F' is provided in *Figure 5—source data 1*.
DOI: https://doi.org/10.7554/eLife.39575.015

The following source data and figure supplement are available for figure 5:

**Source data 1.** Source data for *Figure 5D–F*.
DOI: https://doi.org/10.7554/eLife.39575.017

**Figure supplement 1.** Large volume shifts occur when average impermeant charge is changed via temporary anion species flux across the membrane.
DOI: https://doi.org/10.7554/eLife.39575.016

---

$E_{Cl}$, this results in negligible changes to the driving force for $Cl^-$. This contrasts with the results shown earlier, where adjusting the activity of cation-chloride cotransport modulates both $E_{Cl}$ and the driving force for $Cl^-$ substantially.

## Impermeant anions drive small shifts in chloride driving force by modifying the sodium-potassium-ATPase pump rate under conditions of active chloride cotransport

We next set out to determine how, and under what conditions, the modification of impermeant anions could potentially generate the very small persistent shifts in $Cl^-$ driving force we observed in our models. Due to their small size (<1 mV) these were not detectable during the experimental validation. First, we repeated the simulation performed in *Figure 5A* by changing the mean charge of impermeant anions in the cell, but under conditions where the $Na^+/K^+$-ATPase effective pump rate ($J_p$) was either a cubic function of the transmembrane $Na^+$ gradient (default condition) or was fixed at a constant value (*Figure 6A*). In the case where the pump rate was fixed, adjusting the mean charge of impermeant anions generated no persistent change in $Cl^-$ driving force (*Figure 6A*). Modifying impermeant anions caused a significant change in steady-state intracellular $Na^+$ concentration when $J_p$ was kept constant. However, small shifts in $Cl^-$ driving force occurred only when the effective pump rate was variable, in which minor changes to $[Na^+]_i$ caused significant changes to $J_p$, which in turn resulted in a small shift in $Cl^-$ driving force. There is a direct relationship between the mean charge of impermeant anions (z), $[Na^+]_i$, the effective $Na^+/K^+$-ATPase pump rate and $Cl^-$ driving force. This relationship was abolished when the effective $Na^+/K^+$-ATPase pump rate was held constant by removing its dependence on $Na^+$ (*Figure 6B*). In addition, even large variations in effective pump rate near the default value caused negligible shifts in $Cl^-$ driving force of <1 mV. These results were similar when using the experimentally matched ATPase model by *Hamada et al. (2003)*, with slight differences in final values (*Figure 6—figure supplement 1A*). These small, impermeant anion driven, $Na^+/K^+$-ATPase pump-dependent shifts in $Cl^-$ driving force are completely dependent on the presence of cation-chloride cotransport. In the absence of KCC2, there is no $Cl^-$ driving force as $E_{Cl} = V_m$ (*Figure 6F*).

We then tested whether relaxing the condition of transmembrane osmoneutrality might also alter impermeant anion induced effects on $Cl^-$ driving force. We modeled a situation where increases in

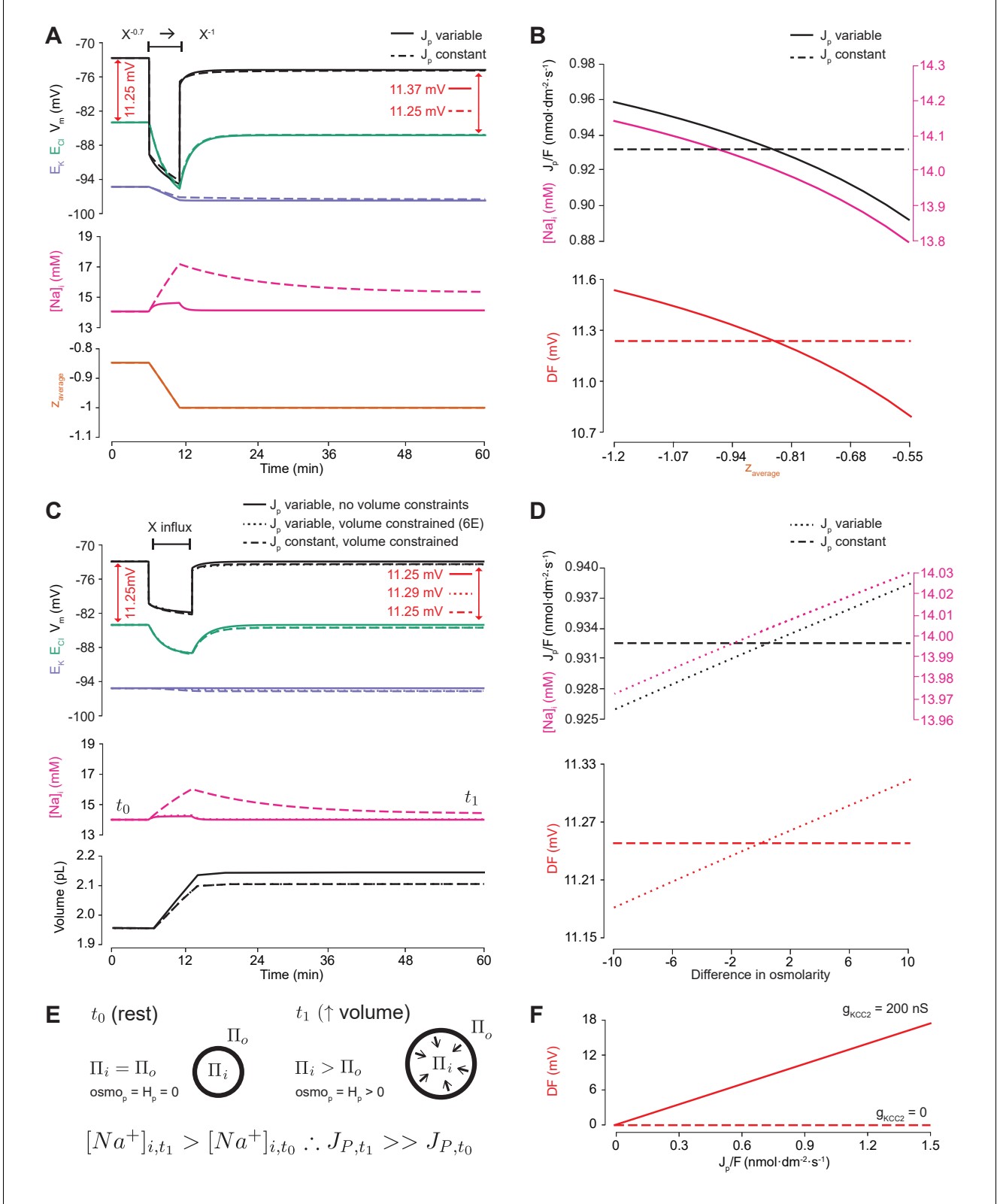

**Figure 6.** Impermeant anions drive small shifts in chloride driving force by modifying the Na$^+$/K$^+$-ATPase pump rate under conditions of active chloride cotransport. (**A**) $E_{Cl}$ (green), $E_K$ (purple) and $V_m$ (black) (top panel), [Na$^+$]$_i$ (pink, middle panel) and mean impermeant anion charge z (orange, lower panel) over time in the default single compartment model. Changing z from −0.85 to −1 generated small, persistent Cl$^-$ driving force (DF) shifts (arrows, red) only when the effective ATPase pump rate (J$_p$) was variable (solid line) and not when J$_p$ was kept constant (dashed line). (**B**) Solving analytically

*Figure 6 continued on next page*

*Figure 6 continued*

across different values of z with either a variable $J_P$ (solid lines) or a constant $J_P$ (dashed line), demonstrates the direct relationship between $Na^+$ (pink), effective pump rate ($J_P$) (top panel) and DF (lower panel, red). (C) $E_{Cl}$ (green), $E_K$ (purple) and $V_m$ (black) (top panel), $[Na^+]_i$ (pink, middle panel) and cell volume (black, lower panel) over time in the default single-compartment model. Impermeant anions of the same charge as the mean charge of the cell were added. A volume constraint was incorporated by adding a hydrostatic force dependent on membrane tension (dashed lines), which resulted in an impermeable anion-induced transmembrane osmotic differential. This caused a small change in DF when $J_P$ was variable (dashed line), but not when $J_P$ was held constant (dotted lines). (D) Solving analytically across osmolarity differences demonstrates the direct relationship between $Na^+$ (pink), effective pump rate ($J_P$) (top panel) and DF (lower panel). Note, the small changes in DF. (E) Schematic explaining the mechanism through which impermeant anion-induced cell swelling in the presence of volume constraints (i.e. membrane tension) result in steady states with equal but non-zero osmotic ($osmo_P$) and hydrostatic pressures ($H_P$), causing transmembrane osmotic differences ($t_1$). This causes small changes in $Na^+$, and hence $J_P$. (F) All $Na^+$/$K^+$-ATPase pump rate-related shifts in the DF require KCC2 activity; in the absence of activity (dashes), no shifts in driving force can occur. In *Figure 6—figure supplement 1*, we show that the results in (A) and (C) are similar when an experimentally-matched model of the $Na^+$/$K^+$-ATPase is used.

DOI: https://doi.org/10.7554/eLife.39575.018

The following figure supplement is available for figure 6:

**Figure supplement 1.** An ATPase model with different kinetics has similar properties to our model's ATPase when average impermeant anion charge in the cell is changed, and is also dependent on the ATPase pump rate's dependence on $Na^+$.

DOI: https://doi.org/10.7554/eLife.39575.019

cell surface area beyond a certain 'resting' surface area generated a hydrostatic pressure (membrane tension), which could balance an osmotic pressure difference of 10 mM between the intra- and extracellular compartments (see *Figure 6C*, schematic in *Figure 6E* and Materials and methods). In this case, adding impermeant anions of default charge z resulted in constrained increases in cell volume, which were accompanied by persistent transmembrane differences in osmolarity and intracellular $Na^+$ concentration. This was sufficient to generate small differences in driving force for $Cl^-$ of <0.2 mV for reasonable increases in cell surface area (*Nichol and Hutter, 1996*; *Dai et al., 1998*). Again, this was entirely due to $Na^+$ driven shifts in the $Na^+$/$K^+$-ATPase effective pump rate. By removing the dependence of $Na^+$/$K^+$-ATPase activity on $Na^+$ concentration, addition of impermeant anions no longer generated persistent shifts in $Cl^-$ driving force (*Figure 6C*). Using the experimentally-matched ATPase model by *Hamada et al. (2003)* generated similar results (*Figure 6—figure supplement 1B*) because the model is also directly dependent on $[Na^+]_i$. We observed a direct relationship between transmembrane osmotic gradient, $[Na^+]_i$, the effective $Na^+$/$K^+$-ATPase pump rate and $Cl^-$ driving force. This relationship was removed when the effective $Na^+$/$K^+$-ATPase pump rate was held constant, with no changes in $Cl^-$ driving force seen despite the generation of the same shift in the transmembrane osmotic gradient (*Figure 6D*).

In summary, changes in $Cl^-$ driving force generated by changing the ionic contributions to cellular charge (by altering the mean charge of impermeant anions) or osmoneutrality (by increasing the contribution of hydrostatic pressure) are due to the alteration of the dynamics of active ion transport mechanisms in the cell. However, these effects are negligible in magnitude and cannot contribute significantly to setting physiologically observed $Cl^-$ driving forces. It is worth reiterating that any non-zero $Cl^-$ driving force is entirely dependent on the presence of active $Cl^-$ cotransport. In our model, in the absence of KCC2, neither the $Na^+$/$K^+$-ATPase nor impermeable anions can shift $Cl^-$ out of equilibrium (*Figure 6F*).

## Changes in cation-chloride cotransport activity generate local differences in chloride reversal and driving force, which depend on cytoplasmic diffusion rates

An important functional question is how $Cl^-$ driving force might be modified at a local level within a neuron. We considered local persistent changes of $Cl^-$ driving force for the case of active transmembrane $Cl^-$ fluxes (*Figure 7*) and impermeant anions (*Figure 8*) by extending the single-compartment model described above into a multi-compartment model or 'virtual dendrite.' This dendrite was 100 μm in length and consisted of 10 compartments, each of 10 μm length and 1 μm diameter. The compartments contained the same mechanisms and default parameterization as the single compartment model described above. Compartmental volume was changed by altering the radius, while holding

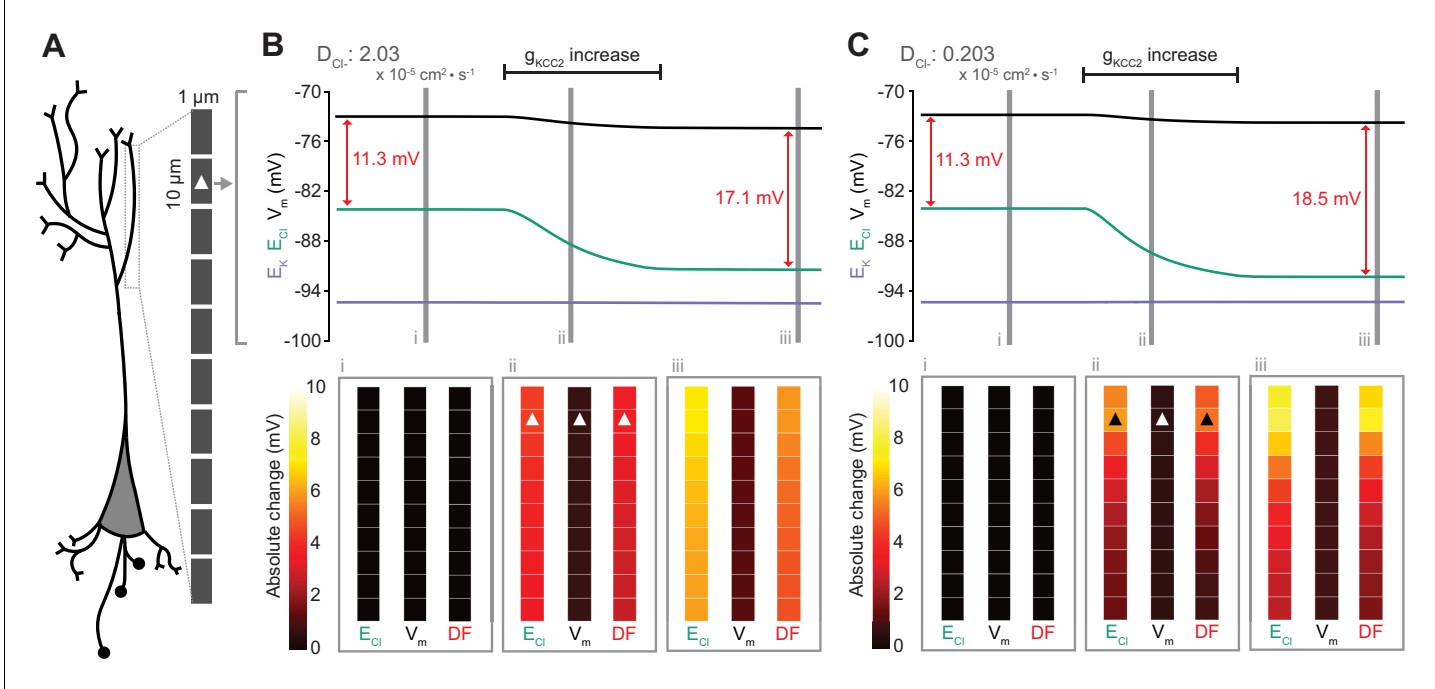

**Figure 7.** Local changes in KCC2 activity generate local differences in chloride reversal and driving force only under conditions of constrained chloride diffusion. (A) Schematic depicting the multi-compartment model representing a virtual dendrite of length 100 μm. The virtual dendrite consists of 10 compartments of length 10 μm and initial radius 0.5 μm. Each compartment contains the same mechanisms and default parameterization as the single compartment model. All ions, except impermeant anions, could move between compartments by electrodiffusion. (B) Top panel, $E_{Cl}$ (green), $E_K$ (purple), $V_m$ (black) and DF (arrows, red) from the second from top compartment (indicated with a white triangle) where the conductance of KCC2 was increased. The insets depict the diameter, and absolute change from baseline of $E_{Cl}$, $V_m$ and DF for all compartments before (i), during (ii) and after (iii) the activity of KCC2 was selectively increased. This resulted in $E_{Cl}$ decreasing in all compartments with minimal changes to $E_K$ and V. Consequently, the Cl⁻ DF (red) increased. In this case the diffusion constant for Cl⁻ ($D_{Cl} = 2.03 \times 10^{-7}$ dm².s⁻¹) resulted in $E_{Cl}$ and DF changes being widespread across the virtual dendrite. (C) Reducing the Cl⁻ diffusion constant to $0.2 \times 10^{-7}$ dm².s⁻¹ resulted in a localized effect of compartment-specific KCC2 activity increases on $E_{Cl}$ and DF.

DOI: https://doi.org/10.7554/eLife.39575.020

the length constant. In addition, all ions, except impermeant anions, could move between compartments by electrodiffusion (*Figure 7A* and Materials and methods).

To explore the local effects of CCC activity, we increased $g_{KCC2}$ from our default value of 20 μS/cm² to 600 μS/cm² in the second distal compartment of the virtual dendrite exclusively. This resulted in a persistent decrease in $E_{Cl}$, concurrent with a modest decrease in $V_m$, resulting in a permanent increase in Cl⁻ driving force and minimal change in compartment volume (*Figure 7B*). The spatial precision of this alteration depended strongly on the diffusion constant for Cl⁻. With a Cl⁻ diffusion constant of $2.03 \times 10^{-7}$ dm².s⁻¹, these alterations spread widely through the virtual dendrite. For example, the change in Cl⁻ driving force was 4.8 mV in the furthermost compartment (90 μm apart) as compared to 5.9 mV in the compartment manipulated. When we decreased the Cl⁻ diffusion constant by one order of magnitude, the change in Cl⁻ driving force was 7.3 mV in the compartment in which KCC2 was adjusted, but only 1.8 mV in the furthermost compartment from the site of manipulation (*Figure 7C*). These findings suggest that local differences in cation-chloride cotransport activity can drive spatially restricted differences in Cl⁻ driving force under conditions of constrained Cl⁻ diffusion; however, under conditions of typical ionic diffusion the effect of Cl⁻ transport by KCC2 is relatively widespread.

## Local impermeant anions do not appreciably affect the local driving force for chloride

Following the last result, we considered whether changing impermeant anions in part of a dendrite could create a local area with a different Cl⁻ driving force compared to the rest of the cell. We first

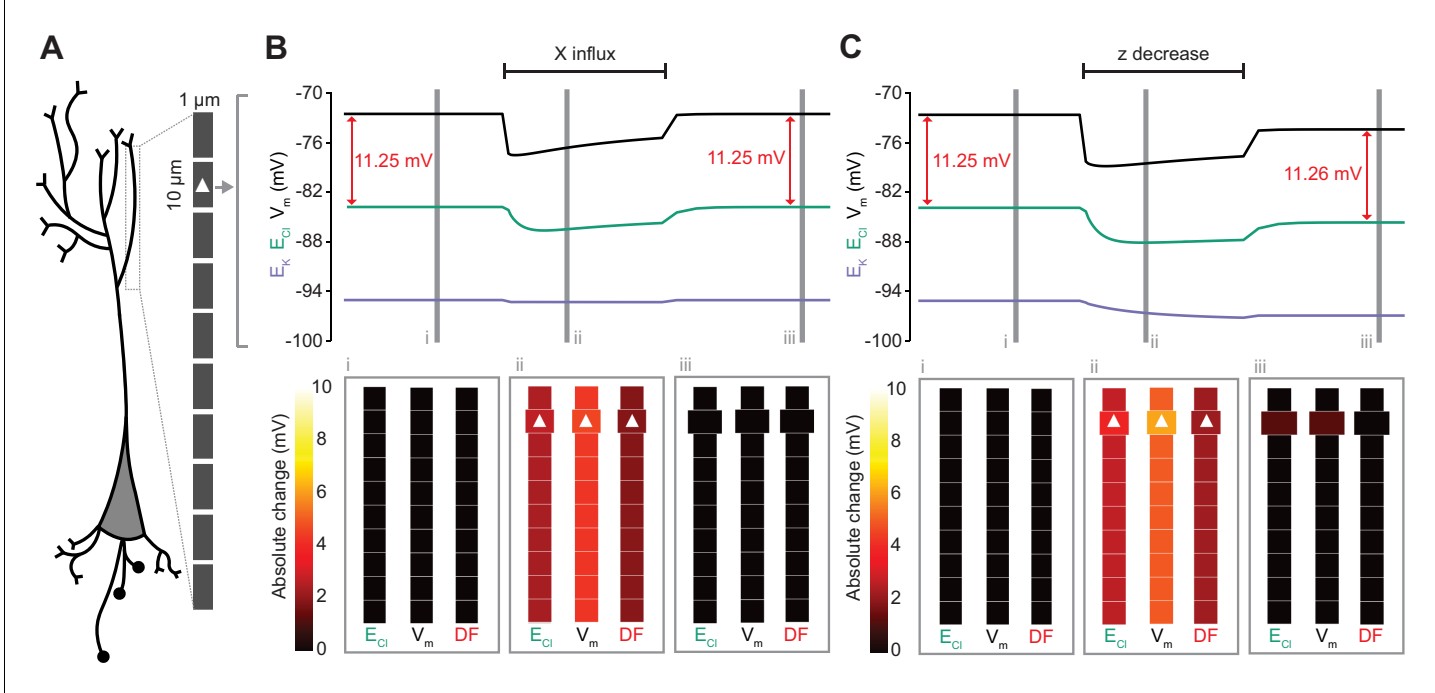

**Figure 8.** Local changes in impermeant anions do not establish the local driving force for chloride. (A) Multi-compartment model of a 10 compartment 100 μm virtual dendrite as in *Figure 7*. (B) Top panel, $E_{Cl}$ (green), $E_K$ (purple), $V_m$ (black) and DF (arrows, red) within the compartment where additional impermeant anions were exclusively added (indicated with a white triangle). The insets depict the diameter, and absolute change from baseline for $E_{Cl}$, $V_m$ and DF for each compartment of the virtual dendrite before (i), during (ii) and after (iii) impermeant anions were added. The selective addition of impermeant anions of default charge (z = −0.85) to the 2nd from top compartment resulted in transient but non-permanent shifts in $E_{Cl}$, $V_m$ and the Cl⁻ DF in all compartments. The volume of the compartment where impermeant anion addition occurred increased permanently. (C) Traces and insets as in 'B' showing the addition of impermeant anions of more negative charge in order to decrease z in the second from top compartment specifically. Note that during addition of impermeant anions, $E_{Cl}$, $E_K$, $V_m$ changed. We also observed persistent decreases in $E_{Cl}$ and $V_m$ in the compartment manipulated, with a negligible change in DF (0.01 mV). Again, impermeant anion addition resulted in an increase in the volume of the specific dendritic compartment manipulated.

DOI: https://doi.org/10.7554/eLife.39575.021

added impermeant anions of the default charge (z = −0.85) exclusively to the second-most distal compartment of the virtual dendrite while measuring the Cl⁻ reversal, $V_m$ and Cl⁻ driving force in all compartments. During addition of the impermeant anions, $E_{Cl}$ and $V_m$ decreased with an accompanying decrease in Cl⁻ driving force. However, following cessation of impermeant anion influx, all parameters returned to baseline levels, except for the volume of that specific compartment, which showed a modest increase (*Figure 8B*). This suggests that local addition of impermeant anions of mean charge has no local effect on Cl⁻ homeostasis but can affect the volume of the compartment concerned.

Next, we again added impermeant anions to the second-last compartment of the virtual dendrite, but this time we added impermeant anions with a more negative charge than (z = −1) than the current mean. This resulted in the mean charge of impermeant anions in that compartment becoming more negative (*Figure 8C*). During the addition of the impermeant anions, $E_{Cl}$ and $V_m$ decreased across the dendrite, but with small accompanying shifts in Cl⁻ driving force. Following cessation of local impermeant anion influx, a persistent shift in $E_{Cl}$ and $V_m$ was observed specifically in the compartment manipulated. However, this generated a negligible, persistent change in Cl⁻ driving force (<0.01 mV change in driving force for a compartment specific change in z from −0.85 to −0.93), only within that specific compartment of the virtual dendrite. Again, impermeant anion addition resulted in a permanent increase in the volume of the compartment concerned. This finding suggests that local impermeant anions can adjust the Cl⁻ reversal potential locally, but are not well-placed to cause significant, permanent shifts in the driving force for Cl⁻. Indeed, electrodiffusion may further limit the degree to which local changes in impermeant ion charge can modify driving forces through

alterations in active ionic transport: the resulting permanent Cl⁻ driving force changes in the multi-compartment model are many times smaller than the shifts in the single compartment version (as compared to *Figure 6B*).

## Discussion

The driving force for Cl⁻ is a fundamental parameter affecting the excitability of neuronal networks (*Raimondo et al., 2017*). Recently, impermeant anions, rather than CCCs, have been suggested as the primary determinants of the neuronal driving force for Cl⁻ (*Glykys et al., 2014*). Here, we have explored the determinants of the Cl⁻ driving force in neurons by deriving theoretical models based on biophysical first principles. We show that the Na⁺/K⁺-ATPase, baseline K⁺, Na⁺ and Cl⁻ conductances, mean charge of impermeant anions, water permeability and CCCs, likely all play roles in setting neuronal $[Cl^-]_i$. However, our findings suggest that while impermeant anions can contribute to setting the $[Cl^-]_i$ in neurons, they can only affect Cl⁻ driving force by modifying the activity of active transport mechanisms (i.e. the Na⁺/K⁺-ATPase). Our modelling and experimental data demonstrate that under physiologically relevant conditions, impermeant anions do not alter the Cl⁻ driving force significantly. In contrast, CCCs are well placed to modulate Cl⁻ driving force and hence inhibitory signaling.

Previous theoretical models, which account for the dynamics of Cl⁻ ions, have been useful in determining how changes to the driving force for Cl⁻ are critical for controlling the effect of synaptic inhibition in the brain (*Qian and Sejnowski, 1990*; *Staley and Proctor, 1999*; *Doyon et al., 2011*; *Jedlicka et al., 2011*; *Lewin et al., 2012*; *Mohapatra et al., 2016*). Whilst these models have included the Na⁺/K⁺-ATPase, the interacting dynamics of several ion species, CCCs (*Doyon et al., 2011*; *Krishnan and Bazhenov, 2011*), electrodiffusion (*Qian and Sejnowski, 1989*) and impermeant anions and volume regulation (*Dijkstra et al., 2016*), none have combined all these mechanisms to explore how their combination determines the local driving force for Cl⁻. Our theoretical approach is based on the pump-leak formulation (*Tosteson and Hoffman, 1960*). It suggests that mammalian cells maintain their volume under osmotic stress generated by impermeant anions and the Donnan effect by employing active transport of Na⁺ and K⁺ using the Na⁺/K⁺-ATPase (*Armstrong, 2003*; *Kay, 2017*). A Donnan equilibrium, a true thermodynamic equilibrium requiring no energy to maintain it, is not possible in cells with pliant membranes like neurons (*Sperelakis, 2012*).

Our model conforms to the pump-leak formulation: abolishing the activity of the Na⁺/K⁺-ATPase leads to cell swelling, progressive membrane depolarization and rundown of ionic gradients, including that of Cl⁻. Therefore, the Na⁺/K⁺ ATPase is a fundamental cellular parameter that stabilizes cell volume and determines all ionic gradients including that of Cl⁻ and hence must be considered in any attempt to model ion homeostasis. Interestingly however, we demonstrate that above a certain level of Na⁺/K⁺-ATPase activity, even many fold changes in pump rate have minimal effects on volume, $E_{Cl}$ and $V_m$. This might explain recent experimental findings in which periods of Na⁺/K⁺ ATPase inhibition using ouabain caused modest changes to cell volume (*Glykys et al., 2014*). It therefore seems unlikely that neurons adjust the Na⁺/K⁺-ATPase as a means for modulating Cl⁻ driving force.

Baseline ion conductances are another important factor that affect Cl⁻ driving force. Our model is consistent with recent experimental results that demonstrate that increased neuronal Na⁺ conductance (for example by activating NMDA receptors, or preventing closure of voltage-gated Na⁺ channels), leads to progressive neuronal swelling, membrane depolarization and Cl⁻ accumulation (*Rungta et al., 2015*) – the primary pathological process in cytotoxic edema (*Liang et al., 2007*). We also show that tonic neuronal Cl⁻ conductance only affects baseline $[Cl^-]_i$ and driving force in the presence of CCCs. Without active Cl⁻ flux, which CCCs provide, there is no driving force for passive Cl⁻ flux and hence no mechanism for $[Cl^-]_i$ changes resulting from selective modification of a Cl⁻ conductance. This is consistent with both classic (*Misgeld et al., 1986*; *Thompson and Gähwiler, 1989*) and recent experimental findings (*Berglund et al., 2016*).

In our model, we find that elevating the activity of KCC2, the most active CCC in mature neurons (*Ben-Ari, 2002*), increases the driving force for Cl⁻ by shifting the reversal potential for Cl⁻ closer to that of K⁺. Interestingly, large shifts (~7 mV) in driving force were associated with very minor (1%) changes in volume or membrane potential. As such, modulating KCC2 represents a specific means for manipulating the neuronal Cl⁻ driving force. This is consistent with traditional dogma, recent and previous experimental results (*Kaila et al., 2014*; *Klein et al., 2018*) as well as our own experimental

validation using furosemide to block the activity of KCC2, which drove significant changes in driving force with little effect on $V_m$. In further support of this, our meta-analysis of numerous experimental studies showed a strong correlation between change in KCC2 expression and Cl⁻ driving force, but not between KCC2 expression and $V_m$. There is an ongoing debate as to whether some cotransporters, including CCCs, might also couple water transport to ion transport (*Zeuthen, 1994*; *MacAulay et al., 2002*; *Gagnon et al., 2004*; *Charron et al., 2006*). Although we did not model the active movement of water by KCC2, this scenario would not alter the central importance of CCCs for setting the Cl⁻ driving force.

Using our multi-compartment model, which incorporated electrodiffusion, we found that local modification of KCC2 activity has a specific local effect on Cl⁻ driving force that is dependent on the characteristics of intracellular Cl⁻ diffusion. Cytoplasmic Cl⁻ diffusion rates had to be reduced substantially before we observed local changes in Cl⁻ driving force driven by KCC2 (*Qian and Sejnowski, 1989*; *Kuner and Augustine, 2000*). Whilst differences in KCC2 activity might generate a gradient in Cl⁻ driving force between large subcellular structures (i.e. dendrites versus soma), our modeling results call into question the idea of synapse-specific regulation of Cl⁻ driving force within the same cellular domain (*Földy et al., 2010*).

*Glykys et al. (2014)* used Cl⁻ imaging and various experimental manipulations to claim that intracellular and extracellular concentrations of impermeant anions ($[X]_i$ and $[X]_o$) set $[Cl^-]_i$ and the Cl⁻ driving force. From our theoretical analysis, we find that modifying the amount of impermeant anions inside or outside neurons has no persistent effect on $[Cl^-]_i$ or Cl⁻ driving force, unless we include a mechanism that allows a transmembrane osmotic pressure differential to develop that indirectly affects active transport mechanisms. Even in this case, under transmembrane pressure differentials that do not lyse the membrane (*Nichol and Hutter, 1996*), Cl⁻ driving force changes are negligible (<1 mV). Recently, it has been suggested that the viscoelastic properties of the cellular cytoskeleton could allow it to take up osmotic shifts created by impermeant anion movement like a sponge (*Sachs and Sivaselvan, 2015*). This would mean that one would not see as large a volume shifts as predicted by our models. In our model, we have assumed that water can pass through the neuronal membrane to equalize osmotic differences. Although it is thought that some neurons do not express aquaporin channels (*Andrew et al., 2007*), water can permeate the phospholipid bilayer (*Fettiplace and Haydon, 1980*). Therefore, whilst differences in neuronal water permeability might affect the time taken to reach steady-state, the steady state values themselves are unlikely to be affected. We conclude that $[Cl^-]_i$ and the Cl⁻ driving force are not determined by the *concentration* of impermeant anions.

However, our theoretical findings offer a potential explanation for recent experimental observations. We show that modifying the mean charge of impermeant anions (i.e. z in $[X^z]_i$), rather than their concentration, can affect $[Cl^-]_i$ and $E_{Cl}$. Relating this to prior experimental observations, *Glykys et al. (2014)* used SYTO64 staining of nucleic acids and perfusion of weak organic acids in conjunction with Cl⁻ imaging to suggest that $[Cl^-]_i$ depends upon internal impermeant anions ($[X]_i$). If such a manipulation modifies the mean charge of internal impermeant anions, and not concentration per se, this could account for the observed changes in $[Cl^-]_i$. Glykys et al. (2014) did not measure $V_m$ or the Cl⁻ driving force in these experiments. The clear prediction from our model is that any manipulation, which changes the mean charge of impermeant anions would not appreciably affect the Cl⁻ driving force because any impermeant anion driven change on $E_{Cl^-}$ is matched by an equivalent effect on $V_m$ due to accompanying shifts in cation concentrations. We have provided experimental support for this prediction by showing that whilst $E_{GABA}$ (and $E_{Cl}$) can be shifted by addition of impermeant anions using electroporation of membrane impermeant anionic dextrans, $V_m$ is shifted in a similar direction resulting in an undetectable change in Cl⁻ driving force. Future experiments could further test our model by electroporating positively charged dextrans which would be predicted to depolarize both $V_m$ and $E_{Cl}$, again with minimal effects on Cl⁻ driving force.

Given prior theoretical predictions (*Kaila et al., 2014*; *Voipio et al., 2014*; *Savtchenko et al., 2017*), it is interesting that our model reveals that changing impermeant anions could affect the Cl⁻ driving force at all. We found that the small (<1 mV) impermeant anion-driven changes in Cl⁻ driving force observed in our model were caused by indirect effects on Na⁺ concentration and hence Na⁺/K⁺-ATPase activity. The impermeant anion-driven changes in Cl⁻ driving force are even smaller in the multi-compartment model (<0.1 mV), in which electrodiffusion allows local changes in Na⁺ to dissipate. When Na⁺/K⁺-ATPase activity was decoupled from the transmembrane Na⁺

gradient, we found that impermeant anions were unable to cause persistent shifts in Cl⁻ driving force as predicted theoretically (*Kaila et al., 2014*; *Voipio et al., 2014*; *Savtchenko et al., 2017*). It is important to note that these small, impermeant anion-Na⁺/K⁺-ATPase-driven shifts in Cl⁻ driving force are dependent on the presence of cation-chloride cotransport in the form of KCC2 and would entail changes in energy use by the Na⁺/K⁺-ATPase. In other words, active transport mechanisms are again required to drive changes in Cl⁻ homeostasis.

In summary, our theoretical models, which are derived from well-established physical principles, are consistent with our own experimental data and that of others (*Glykys et al., 2014*; *Kaila et al., 2014*; *Klein et al., 2018*), and suggest that impermeant anions alone cannot shift Cl⁻ out of equilibrium across the neuronal membrane. Were neurons to alter impermeant anion concentration or charge, the resting membrane potential would be modified with little effect on the Cl⁻ driving force. Our work confirms the central importance of CCC activity in determining the effects of inhibitory synaptic transmission in the nervous system.

## Materials and methods

### Single-compartment model

The single-compartment model consisted of a cylindrical semipermeable membrane separating the extracellular solution from the intracellular milieu with variable volume (*Figure 1* ). The extracellular ionic concentrations were assumed constant (*Table 1*). Permeable ions in the model were K⁺, Na⁺ and Cl⁻ with their usual charges, while impermeant anions X were assumed to be a heterogeneous group of impermeant chemical species with a mean intracellular charge z and a mean extracellular charge -1. The default z (-0.85) was chosen on the basis of known resting intracellular ion concentrations (*Lodish et al., 2009*; *Raimondo et al., 2015*) and osmolarity ($\Pi$). Bicarbonate ions were not included in our model as a permeant anion as they were assumed to be important for acute depolarizing effects (via GABA$_A$Rs) rather than the chronic shifts in Cl⁻ driving force, which are the focus of this work (*Staley and Proctor, 1999*). The model included ionic leak currents for the permeable ions, Na⁺/K⁺-ATPase transporters and a CCC, in this case the K⁺-Cl⁻ cotransporter (KCC2). KCC2 and not NKCC1 is thought to be the most active CCC in mature neurons (*Ben-Ari, 2002*), therefore, to maintain conceptual simplicity only KCC2 was modeled. Cell volume (w) change was based on osmotic water flux and incorporated a membrane surface area scaling mechanism. An analytical solution to the model at steady state was derived using standard techniques and can be found in *Supplementary file 1*. The numerical model was initialized assuming conditions close to electroneutrality and an osmotic equilibrium between the intracellular and extracellular compartments. A forward Euler approach was used to update variables at each time step (dt) of 1 ms. Using a smaller dt did not influence the results in *Figure 1–5*. Code was written in Python 2 and is available on GitHub (*Düsterwald and Currin, 2018*; copy archived at https://github.com/elifesciences-publications/model-of-neuronal-chloride-homeostasis). The GitHub repository includes a file of Supplementary figures, in which we display transmembrane fluxes of all ions and water for relevant simulations. An example figure displaying ionic flux for all ions is available for *Figure 4C* in *Figure 4—figure supplement 1*.

### Membrane potential

The membrane potential V$_m$ was based on the 'Charge Difference' approach (*Rybak et al., 1997*; *Fraser and Huang, 2004*) as follows:

$$V_m = \frac{F\left([Na^+]_i + [K^+]_i - [Cl^-]_i + z[X^z]_i\right)}{C_m A_m} \tag{1}$$

where F is Faraday's constant, C$_m$ is the unit membrane capacitance and A$_m$ is calculated as the ratio of the surface area (of the cylinder) to cell volume. The term in brackets is the sum of all ionic charges within the cell. This approach has the advantage that the initial voltage can be calculated without needing to assume a steady state as is required for by the Goldman-Hodgkin-Katz (GHK) equation.

**Table 1.** Constants, default parameters and usual steady state values for variables used in the biophysical models.

| | Value | Description |
|---|---|---|
| Constants | | |
| F | 96485.33 C/mol | Faraday constant |
| R | 8.31446 J/(K.mol) | Universal gas constant |
| T | 310.15 K | Absolute temperature (=37°C) |
| Parameters | | |
| $C_m$ | $2\times$ - 6 $10^{-6}$ F/cm$^2$ | Unit membrane capacitance (*Qian and Sejnowski, 1989*) |
| $g_{Na}$ | 20 μS/cm$^2$ | Na$^+$ leak conductance (*Kager et al., 2000*) |
| $g_K$ | 70 μS/cm$^2$ | K$^+$ leak conductance (*Kager et al., 2000*) |
| $g_{Cl}$ | 20 μS/cm$^2$ | Cl$^-$ leak conductance |
| $g_{KCC2}$ | 20 μS/cm$^2$ | KCC2 conductance (*Doyon et al., 2016*) |
| $v_w$ | 0.018 dm$^3$/mol | Partial molar volume of water (*Hernández and Cristina, 1998*) |
| $p_w$ | 0.0015 dm/s | Osmotic permeability (*Hernández and Cristina, 1998*) |
| $k_m$ | 25 N/dm | Variable for membrane tension (higher than reported (*Dai et al., 1998*): used in this paper to accentuate differences in osmolarity) |
| p | 0.1 C/(dm$^2$.s) | Default pump rate constant |
| [Na$^+$]$_o$ | 145 mM | Extracellular Na$^+$ concentration |
| [K$^+$]$_o$ | 3.5 mM | Extracellular K$^+$ concentration |
| [Cl$^-$]$_o$ | 119 mM | Extracellular Cl$^-$ concentration |
| [X$^-$]$_o$ | 29.5 mM | Extracellular impermeant anion (X) concentration (*Lodish et al., 2009*; *Raimondo et al., 2015*) |
| $D_{Na}$ | 1.33 × - 7 $10^{-7}$ dm$^2$/s | Na$^+$ diffusion constant (*Hille, 2001*) |
| $D_K$ | 1.96 × - 7 $10^{-7}$ dm$^2$/s | K$^+$ diffusion constant (*Hille, 2001*) |
| $D_{Cl}$ | 2.03 × - 7 $10^{-7}$ dm$^2$/s | Cl$^-$ diffusion constant (*Hille, 2001*) |
| Variables (default steady state) | | |
| $V_m$ | −72.6 mV | Membrane potential |
| [Na$^+$]$_i$ | 14.0 mM | Intracellular Na$^+$ concentration |
| [K$^+$]$_i$ | 122.9 mM | Intracellular K$^+$ concentration |
| [Cl$^-$]$_i$ | 5.2 mM | Intracellular Cl$^-$ concentration |
| [X$^z$]$_i$ | 154.9 mM | Intracellular impermeant anion (X) concentration |
| z | −0.85 | Mean charge of intracellular X (*Lodish et al., 2009*; *Raimondo et al., 2015*) |
| w | 2.0 pL (single compartment) 0.078 pL (multi-compartment) | Volume |

DOI: https://doi.org/10.7554/eLife.39575.022

## Permeable ion concentrations

Intracellular concentrations of the permeable ions Na$^+$, K$^+$ and Cl$^-$ were updated individually by summing trans-membrane fluxes. Leak currents were calculated using Ohm's Law, $I = g(V_m - E)$. In addition, Na$^+$ and K$^+$ were transported actively by the Na$^+$/K$^+$-ATPase, with pump rate J$_p$, which was approximated by a cubic function dependent on the transmembrane sodium gradient, following (*Keener and Sneyd, 1998*):

$$J_p = \mathrm{P}\left(\frac{[Na^+]_i}{[Na^+]_o}\right)^3, \tag{2}$$

where P is the pump rate constant. Because it is a function of the sodium gradient, J$_p$ decreases as [Na$^+$]$_i$ depletes. This formulation has been shown to be similar to more accurate kinetic models reliant on both the Na$^+$ gradient and ATP concentration (*Keener and Sneyd, 1998*). To switch the

ATPase pump on or off (*Figure 1C*), P was decreased/increased exponentially over 10–20 min, consistent with previous reports of the dynamics of inhibition of the ATPase by ouabain and in turn the inhibition of ouabain's effects by potassium canrenoate (*Baker and Willis, 1972*; *Yeh and Lazzara, 1973*). The ATPase pumps 2 $K^+$ ions into the cell for every 3 $Na^+$ ions out and these constants must be multiplied by $J_p$ for each ion, respectively. $K^+$ and $Cl^-$ were also modified by flux through the type 2 K-Cl cotransporter (KCC2), which has a stoichiometry of 1:1 and transports both ions in the same direction. Flux though KCC2, $J_{KCC2}$ (*Doyon et al., 2016*), was modeled as follows:

$$J_{KCC2} = g_{KCC2}(E_K - E_{Cl}),$$ (3)

where $g_{KCC2}$ is a fixed conductance and $E_K$ and $E_{Cl}$ are the Nernst potentials for $K^+$ and $Cl^-$ respectively. $J_{KCC2}$ is 0 when $E_K = E_{Cl}$. The rate of change of the intracellular concentration of the three permeant ions was given by the following equations, with the Nernst potentials for each ion given by $E_{ion} = \frac{RT}{zF} \ln\left(\frac{[ion]_o}{[ion]_i}\right)$, w indicating the cell volume, and $\frac{dw}{dt}$ as described in *Equation 7* (*Figure 1–5,6A, B,7* and *8 or 9* or Equation 9 (*Figure 6C–E*):

$$\frac{d[Na^+]_i}{dt} = -\frac{A_m}{F}\left(g_{Na}(V_m - E_{Na}) + 3J_p\right) - \frac{1}{w}\frac{dw}{dt}[Na^+]_i$$ (4)

$$\frac{d[K^+]_i}{dt} = -\frac{A_m}{F}\left(g_K(V_m - E_K) - 2J_p - J_{KCC2}\right) - \frac{1}{w}\frac{dw}{dt}[K^+]_i$$ (5)

$$\frac{d[Cl^-]_i}{dt} = \frac{A_m}{F}\left(g_{Cl}(V_m - E_{Cl}) + J_{KCC2}\right) - \frac{1}{w}\frac{dw}{dt}[Cl^-]_i.$$ (6)

## Volume

In most calculations, because the osmotic flux of water is expected to be faster than ion fluxes, the volume of the cell (w) was adjusted to reduce the difference between $\Pi_i$ (intracellular osmolarity) and $\Pi_o$ (extracellular osmolarity) at each time step by explicitly modelling water flux, where $v_w$ is the partial molar volume of water, $p_w$ the osmotic permeability of a biological membrane and SA the surface area (*Hernández and Cristina, 1998*):

$$\frac{dw}{dt} = v_w \cdot p_w \cdot SA \cdot (\Pi_i - \Pi_o).$$

In some calculations (*Figure 6C–E*) in which we allowed transmembrane differences in osmolarity to develop, we assumed that at rest the cylindrical cell had a radius of $r_a$ and zero pressure across the membrane, and that the tension (T) in the membrane followed Hooke's law such that the tension was proportional to the difference between the dynamic circumference of the cell and that of the resting state. From Laplace's law the hydrostatic pressure in the cell was given by:

$$H_p = \begin{cases} 4\pi k_m(1 - \frac{r_a}{r}) & r > r_a \\ 0 & , otherwise \end{cases}$$ (8)

where $k_m$ is the spring constant of the membrane (*Sachs and Sivaselvan, 2015*). *Equation 7* was thus reformulated:

$$\frac{dw}{dt} = v_w \cdot p_w \cdot SA \cdot \left(\Pi_i - \Pi_o - \frac{H_p}{RT}\right).$$ (9)

In order to simulate extreme conditions of constrained volume, a larger $k_m$ was employed than is realistic (*Dai et al., 1998*). Intracellular ion concentrations were updated again after volume change at each time step. Volume changes were manifested in the cylindrical compartment as change in the radius. In *Figures 1–6*, the cell was initialised with diameter 10 μm and length 25 μm.

## Anion flux

Impermeant anions were manipulated in the compartment in *Figures 4–6* and *Figure 8* through several mechanisms. Anions could be added to the compartment at a constant rate and have either the

default intracellular X charge z = −0.85 (*Figures 4C*, *6C* and *8B*), or a different charge (*Figures 5C*, *6A* and *8C*). In these cases, the number of moles of X in the compartment was increased. Alternatively, the charge of a species of intracellular X was slowly changed imitating a charge-carrying transmembrane reaction (*Figures 5A* and *6A*). In this case, the number of moles of intracellular X did not change and it was assumed charge imbalance was compensated by the extracellular milieu. Finally, extracellular X⁻ was changed in *Figure 4D* by removing as much Cl⁻ as X⁻ was added, thus maintaining osmolarity and electroneutrality in the extracellular space.

## Multi-compartment model

The single-compartment dendrite model was incorporated in a multi-compartment model by allowing electrodiffusion to occur between individual compartments operating as described above. Compartments were initialised with a radius of 0.5 μm and length of 10 μm. Compartments were linked linearly without branching; 10 connected compartments in total were used. The time step dt was decreased to $10^{-3}$ ms for simulations in multiple compartments. Code was written in Python 3 and is available on GitHub (*Düsterwald and Currin, 2018*).

*Electrodiffusion.* The Nernst-Planck equation (NPE) was used to model one-dimensional electrodiffusion, based on *Qian and Sejnowski (1989)*. The NPE incorporates fluxes because of diffusion and drift (i.e. the movement of ions driven by an electric field). It has been shown to be more accurate than using $J_{diffusion}$ alone in small structures like dendrites (*Qian and Sejnowski, 1989*). The NPE for J the flux density of ion C is calculated as:

$$J = -D\frac{zF^2}{RT}[C]\frac{dV_m}{dx} - DF\frac{d[C]}{dx},$$ (10)

where D is the diffusion constant of ion C (*Table 1*), z is its charge, [C] is its concentration and x is the distance along the longitudinal axis over which electrodiffusion occurs. The NPE was implemented between compartments i and i + 1, assuming the i→i + 1 direction was positive, using a forward Euler approach. The midpoints of the compartments were used to calculate dx, i.e. $dx = \frac{h_i + h_{i+1}}{2}$, and the concentrations of C in each compartment were averaged to obtain $J_{drift}$, ensuring that $J_{i \to i+1} = J_{i+1 \to i}$, where the fluxes had units of mol/(s.dm²):

$$J_{i \to i+1} = -D\left(\frac{zF}{RT}\frac{([C]_i + [C]_{i+1})}{2}\frac{dV_m}{dx} + \frac{d[C]}{dx}\right).$$ (11)

The flux was multiplied by the surface area between compartments and then divided by compartment volume to determine the flux in terms of molar concentration (M/s), i.e. $\frac{\pi r^2}{\pi r^2 h_i} = \frac{1}{h_i}$, and finally implemented numerically with a forward Euler approach. The implementation mirrored that in Qian and Sejnowski (1989) for non-branching dendrites, but was adjusted for compartments whose volumes can change:

$$C_{i \to i+1} = -\frac{dt}{h_i}D\left(\frac{zF}{RT}\frac{([C]_i + [C]_{i+1})}{2}\frac{(V_{m_i} - V_{m_{i+1}})}{dx} + \frac{([C]_i - [C]_{i+1})}{dx}\right).$$ (12)

## Systematic review

A literature search was performed to identify experimental studies that aimed to correlate a change in KCC2 expression with changes in [Cl⁻]ᵢ. The MEDLINE database was used and accessed via the PubMed online platform. Search terms included 'chloride', 'Cl', 'intracellular', 'KCC2', 'cotransporter', 'neuronal', 'GABA' using appropriate Boolean operators. All 26 studies that demonstrated changes in KCC2 expression and $E_{GABA}$ were considered for the meta-analysis. As there is a well-described differential expression of KCC2 and NKCC1 at different stages of development, with KCC2 expression increasing and NKCC1 expression decreasing across development, only studies that used tissue older than postnatal day seven were included (eight included data from younger animals). Other exclusion criteria included: reporting a significant change in NKCC1 (five studies); use of non-rodent tissue (two studies); no quantification of the change in KCC2 (two studies). Nine experiments from eight studies met all criteria and were included (*Coull et al., 2003*; *Lagostena et al., 2010*; *Lee et al., 2011*; *Ferrini et al., 2013*; *Campbell et al., 2015*; *MacKenzie and Maguire, 2015*; *Mahadevan et al., 2015*; *Tang et al., 2015*). However, one study

did not report the change in $V_m$ and hence was excluded from the figure (*Mahadevan et al., 2015*). Data used in regression can be seen in *Supplementary file 2* (Table S2-1) and includes the analysis for regression against change in $[Cl^-]_i$. To accommodate varied experimental preparations and techniques influencing data quality and biases, a 34-point scoring system was designed to weight the studies (Table S2-2). A weighted least squares regression model was then used to correlate the percentage (%) change in KCC2 expression versus change in Cl$^-$d driving force.

## Slice preparation and electrophysiology

For all experiments, organotypic slices were prepared from rodent brain tissue. Wistar rats were used for the experiments testing the effects of CCC blockade. However, to allow for optogenetic manipulation, a crossed mouse strain on a C57BL/6 background was used. This mouse strain was a cross between mice expressing cre-recombinase in glutamic acid decarboxylase 2 (GAD2) positive interneurons (GAD2-IRES-cre, Jax Lab strain 010802) and mice with a loxP-flanked STOP cassette preventing transcription of the red-fluorescent protein tdTomato (a cre-reporter strain, Ai4, Jax Lab strain 007914). This created the GAD2-cre-tdTomato strain which resulted in cre-recombinase and tdTomato expression in all GABAergic interneurons (*Taniguchi et al., 2011*). The use of all animals was approved by either the University of Oxford (rat) or the University of Cape Town (mouse) animal ethics committees.

Organotypic brain slices were prepared using 7 day old Wistar rats (CCC experiment) or crossed GAD2-IRES-cre (RRID:IMSR_JAX:010802) and Ai14 tdtomato reporter (RRID:IMSR_JAX:007914) mice (impermeant anion experiment) and followed the protocol originally described by *Stoppini et al., 1991*. Briefly, brains were extracted and swiftly placed in cold (4°C) dissection media consisting of Gey's Balanced Salt Solution (GBSS #G9779, Sigma-Aldrich, USA) supplemented with D-glucose (#G5767, Sigma-Aldrich, USA). The hemispheres were separated and individual hippocampi were removed and immediately cut into 350 μm slices using a McIlwain tissue chopper (Mickle, UK). Cold dissection media was used to rinse the slices before placing them onto Millicell-CM membranes (#Z354988, Sigma-Aldrich, USA). Slices were maintained in culture medium consisting of 25% (vol/vol) Earle's balanced salt solution (#E2888, Sigma-Aldrich, USA); 49% (vol/vol) minimum essential medium (#M2279, Sigma-Aldrich, USA); 25% (vol/vol) heat-inactivated horse serum (#H1138, Sigma-Aldrich, USA); 1% (vol/vol) B27 (#17504044, Invitrogen, Life Technologies, USA) and 6.2 g/l D-glucose. Slices were incubated in a 5% carbon dioxide ($CO_2$) humidified incubator at between 35–37°C. Recordings were made after 6–14 days in culture. Previous studies have shown that after 7 days in culture (equivalent to postnatal day 14) GABAergic signaling and $E_{GABA}$ has sufficiently developed to a level resembling mature nervous tissue (*Streit et al., 1989*; *Wright et al., 2017*). For the impermeant anion experiment, the mouse organotypic brain slices were injected 1 day after culture with adeno-associated vector serotype 1 (AAV1) containing a double-floxed sequence for channelrhodopsin (ChR2) linked to a yellow fluorescent protein (YFP) tag driven by the elongation factor one promoter (UNC Vector Core, USA). The vector was diffusely injected into slices using a custom-built Openspritzer pressurized ejection device (*Forman et al., 2017*). Slices were left for 6 days in culture to allow for robust expression of ChR2-YFP in GAD2+ interneurons.

For recordings, slices were transferred to a submerged chamber which was perfused with artificial cerebro-spinal fluid (aCSF) bubbled with carbogen gas (95% oxgen:5% carbon dioxide). aCSF was composed of (in mM): NaCl (120); KCl (3); $MgCl_2$ (2); $CaCl_2$ (2); $NaH_2PO_4$ (1.2); $NaHCO_3$ (23); D-Glucose (11) with pH adjusted to be between 7.35–7.40 using 0.1 mM NaOH. Neurons were visualized using a 20x or 60x water-immersion objective (Olympus, Japan) on a BX51WI upright microscope (Olympus, Japan). Widefield images were obtained using a Mightex CCE-B013-U CCD camera. For the optogenetic experiments, we used epifluorescence microscopy to select slices that exhibited strong ChR2-YFP expression throughout the hippocampal region. Micropipettes were prepared from borosilicate glass capillaries with an outer diameter of 1.20 mm and inner diameter of 0.69 mm (Warner Instruments, USA), using a horizontal puller (Sutter, USA). Recordings were made using Axopatch 1D and Axopatch 200B amplifiers and acquired using pClamp9 (Molecular Devices) or WinWCP software (University of Strathclyde, UK).

For the CCC blockade experiment, gramicidin perforated patch recordings (*Kyrozis and Reichling, 1995*) were performed using glass pipettes containing (in mM): 135 KCl, 4 $Na_2ATP$, 0.3 $Na_3GTP$, 2 $MgCl_2$, 10 HEPES and 80 μg/ml gramicidin (Calbiochem; pH 7.38; osmolarity, 290 mosmol/l). After obtaining a cell-attached patch, the gramicidin perforation process was evaluated by

continuously monitoring the decrease in access resistance. Recordings were started once the access resistance had stabilized between 20–80 MΩ, which usually occurred 20–40 min following gigaseal formation. Rupture of the gramicidin patch, referred to as a break through, induces a large influx of the high Cl⁻ internal solution into the cell. This causes a significant and permanent increase in the $E_{GABA}$ at which time recordings would be discarded. $GABA_AR$ activation was achieved by pressure application of muscimol (10 µM, Tocris, UK), a selective $GABA_AR$ agonist, via a picospritzer. To calculate $E_{GABA}$, $GABA_AR$ currents were elicited at different command voltages. These were a series of 10 mV steps above and below a holding voltage of $-60$ mV. Reported membrane potentials were corrected for the voltage drop across the series resistance for each neuron. Holding current (reflecting membrane current) and total current (reflecting membrane current plus the $GABA_AR$-evoked current) were plotted against the corrected holding potential to generate a current-voltage (I-V) curve. Using this graph, the $E_{GABA}$ was defined as the potential where the total current equals the holding current. Some of the data used for calculating $E_{GABA}$ values in *Figure 3E* was also used in a previous study (*Wright et al., 2017*). $V_m$ was defined as the x-intercept of the holding current, and the driving force as the difference between the two. To block CCC function, the KCC2 blocker furosemide (1 mM, Sigma, USA) was applied.

For the impermeant anion experiments, whole-cell recordings were utilized as electroporation consistently ruptured the gramicidin patches. Pipettes were filled with internal solution composed of (in mM): K-Gluconate (120); KCl (10); $Na_2ATP$ (4); NaGTP (0.3); $Na_2$-phosphocreatinine (10) and HEPES (10). To test the effect of introducing impermeant anions, we electroporated the anionic 10 000 MW Dextran Alexa-Flour 488 (Thermo Fisher, USA) using a separate pipette positioned near the soma of the patched cell. This molecule is a hydrophilic polysaccharide, which is both membrane impermeant and highly negatively charged. Pipettes were filled with a 5% dextran solution in phosphate buffered saline and voltage pulses (5–10 of 20 ms duration, 0.5–1 V) were applied using a stimulus isolator. Successful electroporation of the anionic dextran was confirmed visually by observing the cell fill with the fluorescent dye. Electroporation also resulted in immediate membrane depolarization, which recovered over a period of 1–5 mins. $E_{GABA}$, $V_m$ and driving force were calculated during voltage steps in voltage clamp mode (as above) with $GABA_ARs$ activated via endogenous synaptic release of GABA using photo-activation of GAD2+ interneurons expressing ChR2-YFP with 100 ms pulses of blue light using a 470 nm LED (Thorlabs), in the presence of 5 µM CGP-35348 (Torcis Bioscience, UK) to block $GABA_BR$ activation. $E_{GABA}$, $V_m$ and driving force were calculated before and atleast five mins following electroporation to allow for stabilization of $V_m$, $E_{GABA}$ and driving force. The image of the dextran filled cell was acquired using a confocal microscope (LSM510 Meta NLO, Car Zeiss, Jena, Germany). Analysis was performed using custom written scripts in the MATLAB environment (MathWorks). Results are presented as mean ± SEM.

## Acknowledgements

We thank Juha Voipio for critically reading the manuscript and providing helpful comments and suggestions.

## Additional information

### Funding

| Funder | Grant reference number | Author |
| --- | --- | --- |
| Mandela Rhodes Foundation | | Kira Michaela Düsterwald Richard Joseph Burman |
| Deutscher Akademischer Austauschdienst | DAAD-NRF | Christopher Brian Currin |
| National Research Foundation of South Africa | DAAD-NRF | Christopher Brian Currin |
| H2020 European Research Council | ERC Grant Agreement 617670 | Colin J Akerman |
| Newton Fund | | Joseph Valentino Raimondo |

École Polytechnique Fédérale
de Lausanne    Blue Brain Project    Joseph Valentino Raimondo

The funders had no role in study design, data collection and interpretation, or the decision to submit the work for publication.

## Author contributions

Kira M Düsterwald, Conceptualization, Data curation, Software, Formal analysis, Investigation, Visualization, Methodology, Writing—original draft; Christopher B Currin, Data curation, Software, Methodology, Coded the basis of the object-oriented programme for the extension of the pump-leak model to a multi-compartment environment; Richard J Burman, Data curation, Formal analysis, Validation, Investigation, Visualization, Methodology, Performed the anion experiments, and analysed the anion and furosemide experiments and produced their figure parts, Co-produced and performed the search and scoring systems for the meta-analysis; Colin J Akerman, Conceptualization, Data curation, Methodology, Writing—review and editing; Alan R Kay, Conceptualization, Validation, Methodology, Software, Formal analysis, Investigation, Visualization, Writing—review and editing; Joseph V Raimondo, Conceptualization, Resources, Supervision, Funding acquisition, Visualization, Methodology, Writing—original draft, Project administration

## Author ORCIDs

Kira M Düsterwald (iD) http://orcid.org/0000-0003-3217-5326
Christopher B Currin (iD) https://orcid.org/0000-0002-4809-5059
Richard J Burman (iD) https://orcid.org/0000-0003-3107-7871
Colin J Akerman (iD) https://orcid.org/0000-0001-6844-4984
Alan R Kay (iD) https://orcid.org/0000-0002-2820-6188
Joseph V Raimondo (iD) http://orcid.org/0000-0002-8266-3128

## Ethics

Animal experimentation: No experiments were performed on animals prior to humane killing using cervical dislocation at P7, every effort was made to minimise stress prior to killing. The use of rats in the UK (furosemide experiment) were in accordance with regulations from the United Kingdom Home Office Animals (Scientific Procedures) Act. Use of mice (electroporation experiment) at the University of Cape Town was in accordance with South African national guidelines (South African National Standard: The care and use of animals for scientific purposes, 2008) and was approved by the University of Cape Town Animal Ethics Committee protocol no 014/035.

## Decision letter and Author response

Decision letter https://doi.org/10.7554/eLife.39575.029
Author response https://doi.org/10.7554/eLife.39575.030

# Additional files

## Supplementary files

• Supplementary file 1. Analytical solution derivation.
DOI: https://doi.org/10.7554/eLife.39575.023

• Supplementary file 2. Data and scoring methodology for the meta-analysis in Figure 3F.
DOI: https://doi.org/10.7554/eLife.39575.024

• Transparent reporting form
DOI: https://doi.org/10.7554/eLife.39575.025

## Data availability

Code data is available on GitHub (https://github.com/kiradust/model-of-neuronal-chloride-homeostasis; copy archived at https://github.com/elifesciences-publications/model-of-neuronal-chloride-homeostasis). Experimental data in the form of data spreadsheets has been included, and full experimental data is available on Dryad.

The following dataset was generated:

| Author(s) | Year | Dataset title | Dataset URL | Database and Identifier |
|---|---|---|---|---|
| Düsterwald K, Currin C, Burman R, Akerman C, Kay A, Raimondo J | 2018 | Data from: Biophysical models reveal the relative importance of transporter proteins and impermeant anions in chloride homeostasis | http://dx.doi.org/10.5061/dryad.kj1f3v4 | Dryad Digital Repository, 10.5061/dryad.kj1f3v4 |

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
