## [Decision Letter]

[Editors’ note: a previous version of this study was rejected after peer review, but the authors submitted for reconsideration. The first decision letter after peer review is shown below.]

Thank you for submitting your work entitled "Biophysical models reveal the relative importance of transporter proteins and impermeant anions in chloride homeostasis" for consideration by *eLife*. Your article has been reviewed by Kenton Swartz as Reviewing Editor and a Senior Editor, a Reviewing Editor, and three reviewers. The following individuals involved in review of your submission have agreed to reveal their identity: Gilman Toombes (Reviewer #2); Steve Jones (Reviewer #3).

Our decision has been reached after consultation between the reviewers. Based on these discussions and the individual reviews below, we regret to inform you that your work will not be considered further for publication in *eLife*.

All three reviewers enjoyed reading your manuscript and praised your efforts to construct a mathematical model of cytoplasmic chloride regulation in neuronal cells with a focus on the cation-chloride cotransporters and the role of impermeant ions. There was also consensus that models like yours are essential for interpreting and understanding complex physiological experiments. However, the general consensus among the reviewers and Reviewing Editor is that the model would require considerable experimental testing to be appropriate for *eLife*.

*Reviewer #1:*

The current manuscript constructs a mathematical model of cytoplasmic chloride regulation in neuronal cells with a focus on the cation-chloride cotransporters and the role of impermeant ions. The authors give strong motivating factors for why chloride regulation is so important in electrical signaling. Personally, I find it very interesting how as central neurons mature they switch from KCC2 expression to NKCC1 expression (briefly touched on in the metanalysis figure) which then causes the cytoplasmic chloride to drop from ~20 mM to ~5 mM, and chloride channels then switch from being inhibitory rather than excitatory (with exception of medial habenular neurons and most other non-neuronal cell types in the body which do not undergo this switch). I especially like how this manuscript explores several different scenarios using the mathematical model to determine how Cl, membrane potential, cell volume, and other parameters would change under different conditions.

I personally believe that models like this one are essential for interpreting/understanding complex physiological experiments. However, I am concerned by the lack of experimental tests of the model. In Figure 3C it is shown that KCC2 expression does correlate with changes in chloride concentration, but this figure does not represent the level of correspondence that one would like to see for a quantitative model. I would expect to see fitting to time dependent traces recorded from cells in the presence of different pharmacological agents along with chloride dyes, voltage measurements, measurements of cell volume changes etc. The authors argue that there are no good chloride dyes or quantitative measures of chloride concentration, and I agree with this statement (in addition to sodium and potassium measurements); however, there are dyes out there that could be used to at least test some of these ideas in amore qualitative way, and it is possible to measure membrane potential changes (there are many graphs that suggest changes in voltage in the presence of pharmacological agents), and this would significantly strengthen the paper.

I have also become more and more interested in how we treat transporters and channels in these kinds of cell-based mathematical models. This work uses very generic fluxes for these terms, but it would be more more interesting if detailed kinetic models of the NaKATPase or the CCC transporters (if known) could be used to inform how their biophysical regulation impacts the ionic fluxes and then ultimately impacts cellular homeostasis. In many cases these details are not known for transporters because of a lack of the ability to perform detailed patch clamp on transporters, but the best, most detailed models should be used when they can be.

As it stands, I believe that this paper is ideally suited for Journal of Theoretical Biology or a similar journal – I don't even think that Biophysical Journal would be that interested in this manuscript without some connection/validation with experiment. Overall this manuscript lays out a series of predictions about how cellular homeostasis will be perturbed under different conditions, but the lack of experimental validation reduces my enthusiasm for this work.

*Reviewer #2:*

While it is well established that cells use active pumps and passive leak channels to regulate their membrane potential, cytosol composition and volume, it was recently proposed that the concentration and driving potential of Cl^-^ ions might also be regulated by the impermeant ion concentration (Glykys et al., 2014). In this work, the authors use a conventional pump-leak model to examine how the Cl^-^ concentration and driving potential are modulated by pump rates, leak conductances and impermeant anions. They conclude that while impermeant anions can modulate the intra-cellular chloride concentration, their effect on the driving potential is far smaller than the K^+^/Cl^-^ cotransporter (KCC2). In addition, the authors present a meta-analysis linking KCC2 expression to [Cl^-^], and examine the ability of KCC2 and impermeant anions to locally modulate Cl^-^ concentration and driving force.

The paper is clearly written and provides a thorough theoretical analysis of the controversial "impermeable anion" hypothesis (Voipio et al., 2014). However, there are several ways in which the paper could be strengthened:

1) Modelling choices: It would help to explain why particular ions (e.g. bicarbonate) and transporters (e.g. NKCC) did not need to be included model, provide experimental references for the model parameters listed in Table 1, and discuss whether the effects of the cytoskeleton on osmotic pressure (Sachs and Sivaselvan, 2015), coupled water and ion transport (Delpire and Staley, 2014), or other mechanisms could change the paper's key conclusions.

2) Mechanistic explanation: The idea that changing [A^-^] must impact [Cl^-^] and thus the Cl^-^ driving force is simple and appealing. The authors convincingly demonstrate issues with this "impermeant anion hypothesis", but do not explicitly illustrate how the flow of K^+^, Na+, and water effectively decouples changes in [A^-^] from the Cl^-^ driving force. To help readers, the authors could display the concentrations, potentials and net fluxes of all ions (including Na^+^ and water) in time-dependent simulations (e.g. Figure 1C, Figure 3A, Figure 4A,C,D, Figure 5A/5C, Figure 6A/6C). For example, in Figure 5A, the increasing impermeant anion charge lowers the membrane potential which then drives a larger inflow of K^+^ and Na^+^ than the outflow of Cl^-^ (because g_Cl_ < g_K_ + g_Na_). This clearly conflicts with the assumption of the "impermeant anion hypothesis" that [A^-^] + [Cl^-^] is constant, since most of change in A- is compensated by cations.

3) Experimental Testing: The study would be even more valuable if the authors discussed potential experimental tests of their conclusions. For example, how could one distinguish to what extent cells modulate CCCs (g_KCC2_) rather than the chloride leak conductance (g_Cl_)?

*Reviewer #3:*

This paper presents a pump-leak model including ion conductances and transporters to explore the basis of chloride gradients in cells with KCC2 chloride-potassium transporter. The model is simple (in the most basic case, analytically solvable) yet sufficient to produce realistic ion gradients and voltages. It concludes that KCC2 plays a critical role, and impermeant ions have interesting but quite minor effects (as expected from equilibrium considerations but opposed to a controversial Science paper). This will be of interest both for those interested in how neurons produce chloride gradients (which can vary substantially physiologically), and for broader questions of maintenance of ion homeostasis.

I should admit that I am not an expert in the specific modeling methodology used in this paper, but it looks good as far as I can tell. I have no serious concerns.

[Editors’ note: what now follows is the decision letter after the authors submitted for further consideration.]

Thank you for resubmitting your work entitled "Biophysical models reveal the relative importance of transporter proteins and impermeant anions in chloride homeostasis" for further consideration at *eLife*. Your revised article has been favorably evaluated by Richard Aldrich (Senior Editor), Kenton Swartz (Reviewing Editor), and three reviewers.

This manuscript describes thorough mathematical modelling of the physical mechanisms controlling chloride homeostasis. Concerns raised in an earlier review have been thoroughly addressed, and the additional experiments nicely complement the authors' model. We think this is a careful and valuable study, and at this point our main concern would be the need for clearer explanations of the experimental design and controls. The following are issues the authors should address in revision.

1) Why doesn't E_K_ change in Figure 2 panels? Is it overwhelmed by the Na/K-ATPase flux, and the transporter is set to an E_K_ of -100 mV (Addressed in Figure 3H?)? Actually, in the caption for Figure 2B you say that E_K_ does change:

"Increasing Na^+^ conductance g_Na_ resulted in a progressive increase in steady state E_Cl_, E_K_ and V_m_ with accompanying cell swelling"

However, E_K_ does not change. You do not show E_Na_, but is it also being pegged at about +40 mV for all g_Na_ values?

2) In Figure 3D, can't you just subtract off the drug sensitive current so that it passes through zero to see the reversal potential? We had to look at this way of plotting the currents several times before we got it. Labels near the grey/black/pink curves may help too. Does this value of ~ 4 mV in Figure 3D seem small given the theory predictions in Figure 3B? Does Figure 3H belong in this figure? It isn't clear that it should be here, unless we didn't read this carefully.

3) In subsection “Altering the concentration of intracellular or extracellular impermeant anions, without changing the average charge of impermeant anions, does not affect the steady state gradient or driving force for chloride”, define average charge. It is confusing. In subsection “Changing the average charge of impermeant anions can drive substantial shifts in the reversal potential for chloride, but has negligible effects on chloride driving force”, what is the physical reason why changes to impermeant anions have different effects on Cl^-^ than changing the average charge?

4) Data in Figure 5E, F nicely shows that this manipulation of adding the charged dextrans gives an undetectable shift in the Cl^-^ driving force. That said in Figure 5F some of the DF changes are pretty big (negative) but is offset from others that are positive giving an average that is closer to zero. Not sure what to think about this.

5) For the impermeant anion experiment (Figure 5D-F), the authors have elegantly confirmed the addition of impermeable anions to each neuron via fluorescence microscopy. However, fluorescence microscopy can detect very low concentrations of fluorophores, and so the observation of fluorescence from a neuron might not ensure there has been a meaningful change in average anion charge. Furthermore, while the shift in V_m_ is certainly consistent with an increase in average anion charge, it would be nice to discuss, or better still, experimentally exclude any unintended effects of electroporation. For example, the author's model seems to predict there would be no change in V_m_ after adding a neutral (e.g. dextran-Texas Red) polymer, and that neurons would depolarize after adding a cationic (e.g. FITC-DEAE-Dextran) polymer.

6) For the furosemide experiments (Figure 3D-F), the authors have previously (Wright et al., 2017) described a "within cell" approach in which each neuron serves as its own control. If this study is using this same approach, it would be good to report these controls (e.g. V_m_, E_GABA_ and DF 5 or 15 minute prior to addition of furosemide) to quantify how precisely changes in voltage can be measured. These controls would be especially valuable as the measurements appear to be made in a challenging regime where the access resistance is comparable to the membrane resistance. For example, in Figure 3D, the series resistance (~ 60MOhms baseline, ~ 40 MOhms +Furosemide) is comparable to the resting membrane+leak resistance (~120 MOhms), and so the series resistance correction for E_GABA_ (~ +9mV baseline, +6mV + Furosemide) is comparable to the driving force.

---

## [Author Response]

[Editors’ note: the author responses to the first round of peer review follow.]

Reviewer #1:The current manuscript constructs a mathematical model of cytoplasmic chloride regulation in neuronal cells with a focus on the cation-chloride cotransporters and the role of impermeant ions. The authors give strong motivating factors for why chloride regulation is so important in electrical signaling. Personally, I find it very interesting how as central neurons mature they switch from KCC2 expression to NKCC1 expression (briefly touched on in the metanalysis figure) which then causes the cytoplasmic chloride to drop from ~20 mM to ~5 mM, and chloride channels then switch from being inhibitory rather than excitatory (with exception of medial habenular neurons and most other non-neuronal cell types in the body which do not undergo this switch). I especially like how this manuscript explores several different scenarios using the mathematical model to determine how Cl, membrane potential, cell volume, and other parameters would change under different conditions.

We are encouraged that the reviewer believes that “models like this one are essential for interpreting/understanding complex physiological experiments” and that they “especially like how this manuscript explores several different scenarios using the mathematical model to determine how Cl, membrane potential, cell volume, and other parameters would change under different conditions.” The major critique of the reviewer is that they are “concerned by the lack of experimental tests of the model” and that “the lack of experimental validation reduces my enthusiasm for this work”. We now provide direct experimental validation of our model by confirming its two main and fundamental predictions. We respond to the reviewers comments below.

We also find the switch from NKCC1- to KCC2-predominant expression in neurons very interesting and would like to explore it in greater detail in the future. However, this is not the focus of this current manuscript.

I personally believe that models like this one are essential for interpreting/understanding complex physiological experiments. However, I am concerned by the lack of experimental tests of the model. In Figure 3C it is shown that KCC2 expression does correlate with changes in chloride concentration, but this figure does not represent the level of correspondence that one would like to see for a quantitative model. I would expect to see fitting to time dependent traces recorded from cells in the presence of different pharmacological agents along with chloride dyes, voltage measurements, measurements of cell volume changes etc. The authors argue that there are no good chloride dyes or quantitative measures of chloride concentration, and I agree with this statement (in addition to sodium and potassium measurements); however, there are dyes out there that could be used to at least test some of these ideas in amore qualitative way, and it is possible to measure membrane potential changes (there are many graphs that suggest changes in voltage in the presence of pharmacological agents), and this would significantly strengthen the paper.

We agree that experimental validation would significantly strengthen the paper. The revised manuscript now incorporates experimental evidence which supports the two major predictions of the work. (1) We show that cation-chloride cotransporter (CCC) blockade shifts (E_Cl_) with little change to membrane potential resulting in substantial changes to Cl^-^ driving force (Figure 3). (2) Using a novel electrophysiological method, (electroporation of impermeant anions), we now provide experimental evidence that increasing the average charge of impermeant anions can cause persistent decreases in E_Cl_ and membrane potential but is associated with negligible changes in Cl^-^ driving force (Figure 5).

I have also become more and more interested in how we treat transporters and channels in these kinds of cell-based mathematical models. This work uses very generic fluxes for these terms, but it would be more more interesting if detailed kinetic models of the NaKATPase or the CCC transporters (if known) could be used to inform how their biophysical regulation impacts the ionic fluxes and then ultimately impacts cellular homeostasis. In many cases these details are not known for transporters because of a lack of the ability to perform detailed patch clamp on transporters, but the best, most detailed models should be used when they can be.

More detailed, validated models of the transporters as known to us have now been utilised and the results are included as figure supplements (replications of Figure 1D, Figure 3A, Figure 4C, Figure 5A in the case of the ATPase model, and of Figure 3A in the case of the KCC2 model). We find that the main conclusions of our paper are not changed through employing these more detailed models and therefore we have chosen to retain our original models in the main paper: their relative mathematical simplicity has the advantage of having an analytical solution. However, a future avenue may be to evaluate the impact of these types of models on timedependent fluxes.

As it stands, I believe that this paper is ideally suited for Journal of Theoretical Biology or a similar journal – I don't even think that Biophysical Journal would be that interested in this manuscript without some connection/validation with experiment. Overall this manuscript lays out a series of predictions about how cellular homeostasis will be perturbed under different conditions, but the lack of experimental validation reduces my enthusiasm for this work.

We hope the addition of experimental validation combined with the importance of this issue (i.e. the cellular determinants of Cl^-^ driving force) within the neuroscience field, will increase the reviewer’s enthusiasm for our work.

Reviewer #2:While it is well established that cells use active pumps and passive leak channels to regulate their membrane potential, cytosol composition and volume, it was recently proposed that the concentration and driving potential of Cl^-^ ions might also be regulated by the impermeant ion concentration (Glykys et al., 2014). In this work, the authors use a conventional pump-leak model to examine how the Cl^-^ concentration and driving potential are modulated by pump rates, leak conductances and impermeant anions. They conclude that while impermeant anions can modulate the intra-cellular chloride concentration, their effect on the driving potential is far smaller than the K^+^/Cl^-^ cotransporter (KCC2). In addition, the authors present a meta-analysis linking KCC2 expression to [Cl^-^], and examine the ability of KCC2 and impermeant anions to locally modulate Cl^-^ concentration and driving force.The paper is clearly written and provides a thorough theoretical analysis of the controversial "impermeable anion" hypothesis (Voipio et al., 2014).

We are delighted that the reviewer feels that our “paper is clearly written and provides a thorough theoretical analysis of the controversial "impermeable anion" hypothesis”. We address the reviewer’s comments below.

However, there are several ways in which the paper could be strengthened:1) Modelling choices: It would help to explain why particular ions (e.g. bicarbonate) and transporters (e.g. NKCC) did not need to be included model, provide experimental references for the model parameters listed in Table 1, and discuss whether the effects of the cytoskeleton on osmotic pressure (Sachs and Sivaselvan, 2015), coupled water and ion transport (Delpire and Staley, 2014), or other mechanisms could change the paper's key conclusions.

We thank the reviewer for these excellent suggestions. We now include explanations for why we did not include NKCC1 or bicarbonate in the paper (see Materials and methods section). We also provide references for the parameters in Table 1. We sought to make the model as comprehensive as possible whilst retaining sufficient simplicity to probe the respective roles of CCCs and impermeant anions on Cl^-^ driving force. Further complicating the model would preclude us from finding an analytical solution and we believe this would not affect our ability to compare the respective roles of CCCs and impermeant anions. However, we acknowledge that incorporating bicarbonate ions and pH would be a useful avenue to explore in future. In addition, as suggested by the reviewer, we have now added sections to the discussion where we refer to effects of the “cytoskeleton on osmotic pressure” as well as “coupled ion and water transport”.

2) Mechanistic explanation: The idea that changing [A^-^] must impact [Cl^-^] and thus the Cl^-^ driving force is simple and appealing. The authors convincingly demonstrate issues with this "impermeant anion hypothesis", but do not explicitly illustrate how the flow of K^+^, Na+, and water effectively decouples changes in [A^-^] from the Cl^-^ driving force. To help readers, the authors could display the concentrations, potentials and net fluxes of all ions (including Na^+^ and water) in time-dependent simulations (e.g. Figure 1C, Figure 3A, Figure 4A,C,D, Figure 5A/5C, Figure 6A/6C). For example, in Figure 5A, the increasing impermeant anion charge lowers the membrane potential which then drives a larger inflow of K^+^ and Na^+^ than the outflow of Cl^-^ (because g_Cl_ < g_K_ + g_Na_). This clearly conflicts with the assumption of the "impermeant anion hypothesis" that [A^-^] + [Cl^-^] is constant, since most of change in A- is compensated by cations.

We are encouraged that the reviewer believes that we “convincingly demonstrate issues with this impermeant anion hypothesis". We agree that showing the flux of other ions will illustrate “how the flow of K^+^, Na+, and water effectively decouples changes in [A^-^] from the Cl^-^ driving force”. We have added a supplementary figure with detailed fluxes for all the ions as a Figure supplement to Figure 4, which clearly demonstrates this. We also make this point in the discussion. In addition, we now reference a prepared Jupyter notebook, which can be viewed on our GitHub repository to visualise all of the ion dynamics for all the figures. However, we believe that displaying all the net fluxes for all the ions for almost all the figures would make them impossibly busy and very difficult to follow, especially as we would like the main focus to be on Cl^-^ and impermeant anions.

*3) Experimental Testing: The study would be even more valuable if the authors discussed potential experimental tests of their conclusions. For example, how could one distinguish to what extent cells modulate CCCs (*g_KCC2_*) rather than the chloride leak conductance (*g_Cl_*)?*

We agree that experimental validation would significantly strengthen the paper. We have now included two new experiments in the manuscript, which verify our main predictions: i.e. (1) that modulating CCCs has a significant effect on E_Cl_ and Cl^-^ driving force with minimal effect on V_m_ and (2) that changing the average charge of impermeant anions by addition of impermeant anions can shift both E_Cl_ and V_m_, but has negligible effect on Cl^-^ driving force.

Reviewer #3:This paper presents a pump-leak model including ion conductances and transporters to explore the basis of chloride gradients in cells with KCC2 chloride-potassium transporter. The model is simple (in the most basic case, analytically solvable) yet sufficient to produce realistic ion gradients and voltages. It concludes that KCC2 plays a critical role, and impermeant ions have interesting but quite minor effects (as expected from equilibrium considerations but opposed to a controversial Science paper). This will be of interest both for those interested in how neurons produce chloride gradients (which can vary substantially physiologically), and for broader questions of maintenance of ion homeostasis.I should admit that I am not an expert in the specific modeling methodology used in this paper, but it looks good as far as I can tell. I have no serious concerns.

We thank the reviewer and are encouraged by the reviewer’s comment that “This will be of interest both for those interested in how neurons produce chloride gradients (which can vary substantially physiologically), and for broader questions of maintenance of ion homeostasis.”

[Editors' note: the author responses to the re-review follow.]

This manuscript describes thorough mathematical modelling of the physical mechanisms controlling chloride homeostasis. Concerns raised in an earlier review have been thoroughly addressed, and the additional experiments nicely complement the authors' model. We think this is a careful and valuable study, and at this point our main concern would be the need for clearer explanations of the experimental design and controls. The following are issues the authors should address in revision.1) Why doesn't E_K_ change in Figure 2 panels? Is it overwhelmed by the Na/K-ATPase flux, and the transporter is set to an E_K_ of -100 mV (Addressed in Figure 3H?)? Actually, in the caption for Figure 2B you say that E_K_ does change:"Increasing Na^+^ conductance g_Na_ resulted in a progressive increase in steady state E_Cl_, E_K_ and Vm with accompanying cell swelling"However, E_K_ does not change. You do not show E_Na_, but is it also being pegged at about +40 mV for all g_Na_ values?

We apologise for the misleading description. Neither E_K_ nor E_Na_ are pegged in this figure, nor for the models used for any figure in the manuscript. Intracellular K^+^, Na^+^ and Cl^-^ remain dynamic and are determined by multiple interacting mechanisms (Na^+^/K^+^-ATPase flux, mean charge of impermeant anions, KCC2 flux and flux through ion specific passive conductances). Although it is not obvious (as the changes are small) E_K_ does in fact increase in Figure 2A-C (in Figure 2D, there are no changes in any ionic potentials – although these are not “pegged” – because KCC2 is inactive while non-zero g_Cl_ is altered). In Figure 2B, E_Na_ is not pegged at +40 mV for all g_Na_, but is variable. Indeed, with increasing g_Na_, intracellular Na^+^ concentration increases and E_Na_ decreases. Because the Na^+^/K^+^-ATPase pump rate is a function of the cube of [Na^+^]_i_/[Na^+^]_o_, the increase in [Na^+^]_i_ increases both K^+^ import and Na^+^ export by the Na^+^/K^+^-ATPase. This is why the E_K_ increases are so negligible and the E_Na_ decrease smaller than one might expect with such large changes in g_Na_. I.e the increased active Na^+^/K^+^-ATPase flux compensates for the increased passive K^+^ efflux due to membrane depolarisation. To make this more clear, we have added a figure supplement (Figure 2 —figure supplement 2) in which for varying g_Na_ we plot, E_Na_, E_K_, K^+^ flux through passive channels, K^+^ flux through KCC2 and K^+^ flux through the Na^+^/K^+^-ATPase. We have also amended the legend for Figure 2B to state that increasing Na^+^ conductance leads to minimal increases in E_K_.

2) In Figure 3D, can't you just subtract off the drug sensitive current so that it passes through zero to see the reversal potential? We had to look at this way of plotting the currents several times before we got it. Labels near the grey/black/pink curves may help too. Does this value of ~ 4 mV in Figure 3D seem small given the theory predictions in Figure 3B? Does Figure 3H belong in this figure? It isn't clear that it should be here, unless we didn't read this carefully.

We have amended the confusing presentation of this figure, including the colour scheme. Now we use grey (holding current or membrane current) and black (total current, reflecting membrane current plus muscimol-evoked current) lines on the inset traces to show how these were measured. We have also added labels as suggested. Unfortunately, in Figure 3D, one can’t “subtract off the drug sensitive current (in this case the muscimol mediated current) so that it passes through zero to see the reversal potential”. The reason for this is that although the currents were elicited at the same command voltages (i.e. pre- and post-muscimol puff), because of the series or access resistance, different total currents at each point in time meant that the actual membrane voltages achieved were different (due to a different voltage drop across the series resistance). One will notice that the circles in Figure 3D representing the measured values for the currents (total current in black and holding current in grey) for each voltage step correspond to differing membrane voltages on the x-axis. As a result, a subtraction cannot easily be achieved and the most accurate means of determining the GABA reversal was to observe where the two curves (grey and black) intersect. This is why we have plotted the data in this fashion.

We agree that the example of a ~4 mV shift in driving force after blockade of CCCs is on the small side. However, this is a single example and the mean shift in driving force for the population, as shown in the text and Figure 3E, is larger at 7.3 mV. This seems reasonable to us given that we may not always have achieved complete block of the CCCs and that the expression / activity of CCCs in this system is not accurately characterised. The direction and magnitude of the changes are broadly consistent with our predictions.

We agree with the reviewers comment that Figure 3H is somewhat “out on a limb” in this figure. We have therefore moved it to become a figure supplement for Figure 1 (Figure —figure supplement 2).

3) In subsection “Altering the concentration of intracellular or extracellular impermeant anions, without changing the average charge of impermeant anions, does not affect the steady state gradient or driving force for chloride”, define average charge. It is confusing.

We apologise for the confusion our naming convention caused. In the revised manuscript we no longer refer to z as “the average charge of impermeant anions”, but rather as the “mean charge of impermeant anions”. We now include a clearer explanation and definition In subsection “Altering the concentration of intracellular or extracellular impermeant anions, without changing the average charge of impermeant anions, does not affect the steady state gradient or driving force for chloride”. (“The mean charge (z) is the mean charge of all the different species of impermeant ions in the cell, where charge is the difference between the number of protons and electrons of an ion. Impermeant anions are more abundant than impermeant cations, and so in this manuscript we often refer to the group as impermeant anions rather than ions. For example, were there α impermeant anions of charge -1 and β impermeant anions of charge -2; then z would be -(α+2β)2.”).

In subsection “Changing the average charge of impermeant anions can drive substantial shifts in the reversal potential for chloride, but has negligible effects on chloride driving force”, what is the physical reason why changes to impermeant anions have different effects on Cl^-^ than changing the average charge?

As we demonstrate in Figure 4 and Figure 5, changing the concentration of impermeant anions without altering their mean charge has different effects to a manipulation which does change the mean charge of impermeants. Multiple physical mechanisms interact. When one adds impermeant anions to a cell without disturbing the mean charge of impermeants, the effect is to change the osmotic balance across the membrane, requiring cell swelling to occur in order for steady state / osmotic balance to be restored. The steady state concentrations of all ions remains stable due to the pump-leak mechanism, therefore E_Cl_ and V_m_ do not experience a persistent shift. However, if one changes the mean charge of impermeant ions, the distribution of permeant ions shifts in order to ensure bulk electroneutrality within the cell. As a result, at steady state both E_Cl_ and V_m_ shift to new values along with volume changes.

4) Data in Figure 5E, F nicely shows that this manipulation of adding the charged dextrans gives an undetectable shift in the Cl^-^ driving force. That said in Figure 5F some of the DF changes are pretty big (negative) but is offset from others that are positive giving an average that is closer to zero. Not sure what to think about this.

The reviewers are quite correct that there is certainly variability in the data, which in an experiment of this nature cannot be avoided. We believe that the mean of the population is the most useful metric and supports our model’s predictions.

5) For the impermeant anion experiment (Figure 5D-F), the authors have elegantly confirmed the addition of impermeable anions to each neuron via fluorescence microscopy. However, fluorescence microscopy can detect very low concentrations of fluorophores, and so the observation of fluorescence from a neuron might not ensure there has been a meaningful change in average anion charge. Furthermore, while the shift in V_m_ is certainly consistent with an increase in average anion charge, it would be nice to discuss, or better still, experimentally exclude any unintended effects of electroporation. For example, the author's model seems to predict there would be no change in V_m_ after adding a neutral (e.g. dextran-Texas Red) polymer, and that neurons would depolarize after adding a cationic (e.g. FITC-DEAE-Dextran) polymer.

We are very encouraged by the reviewers’ positive response to this experiment. The reviewers are correct that fluorescence microscopy can detect low concentrations of fluorophores. As the reviewers acknowledge, the shift in V_m_ we observe is consistent with a decrease in mean charge of impermeant ion (i.e. average anion charge). In the discussion, we now discuss future possible experiments along the lines of the reviewers’ excellent suggestions, which would further corroborate the predictions of the model and exclude any unintended effects of the electroporation. Due to capacity constraints and the extended length of time required to perform additional experiments, which we and the reviewers do not deem essential at this point, we have not provided additional experimental data along these line in the revised manuscript.

6) For the furosemide experiments (Figure 3D-F), the authors have previously (Wright et al., 2017) described a "within cell" approach in which each neuron serves as its own control. If this study is using this same approach, it would be good to report these controls (e.g. V_m_, E_GABA_ and DF 5 or 15 minute prior to addition of furosemide) to quantify how precisely changes in voltage can be measured. These controls would be especially valuable as the measurements appear to be made in a challenging regime where the access resistance is comparable to the membrane resistance. For example, in Figure 3D, the series resistance (~ 60MOhms baseline, ~ 40 MOhms +Furosemide) is comparable to the resting membrane+leak resistance (~120 MOhms), and so the series resistance correction for E_GABA_ (~ +9mV baseline, +6mV + Furosemide) is comparable to the driving force.

This experiment used the same “within cell” approach as Wright et al., 2017. On the reviewers’ request we now also report the data for the time point 5 minutes before the “baseline” measure in the text for V_m_, E_GABA_ and DF. The reviewers are quite correct that this is a challenging regime as access/series resistance during gramicidin perforated patches are typically between ~30 MOhms and ~60 MOhms – this necessitated the use of offline series resistance correction. We agree that this correction is comparable to the driving forces measured. However, we do believe that our approach, particularly in the context of the population mean, was able to detect furosemide induced changes in chloride driving force of the magnitude we report.